# Hyperoxia causes miR-34a-mediated injury via angiopoietin-1 in neonatal lungs

Mansoor Syed[1,2,8], Pragnya Das[2], Aishwarya Pawar[2], Zubair H. Aghai[3], Anu Kaskinen[4], Zhen W. Zhuang[5], Namasivayam Ambalavanan [6], Gloria Pryhuber[7], Sture Andersson[4] & Vineet Bhandari [1,2]

Hyperoxia-induced acute lung injury (HALI) is a key contributor to the pathogenesis of bronchopulmonary dysplasia (BPD) in neonates, for which no specific preventive or therapeutic agent is available. Here we show that lung micro-RNA (miR)-34a levels are significantly increased in lungs of neonatal mice exposed to hyperoxia. Deletion or inhibition of miR-34a improves the pulmonary phenotype and BPD-associated pulmonary arterial hypertension (PAH) in BPD mouse models, which, conversely, is worsened by miR-34a overexpression. Administration of angiopoietin-1, which is one of the downstream targets of miR34a, is able to ameliorate the BPD pulmonary and PAH phenotypes. Using three independent cohorts of human samples, we show that miR-34a expression is increased in type 2 alveolar epithelial cells in neonates with respiratory distress syndrome and BPD. Our data suggest that pharmacologic miR-34a inhibition may be a therapeutic option to prevent or ameliorate HALI/BPD in neonates.

[1] Division of Perinatal Medicine, Department of Pediatrics Yale University School of Medicine, New Haven, CT 06510, USA. [2] Section of Neonatology, Department of Pediatrics Drexel University College of Medicine, Philadelphia, PA 19102, USA. [3] Section of Neonatology, Department of Pediatrics Thomas Jefferson University, Philadelphia, PA 19107, USA. [4] Children's Hospital, University of Helsinki and Helsinki University Hospital Helsinki, Helsinki, 00029, Finland. [5] Section of Cardiovascular Medicine, Department of Medicine Yale University School of Medicine, New Haven, CT 06510, USA. [6] Division of Neonatology, Department of Pediatrics University of Alabama at Birmingham, Birmingham, AL 35249, USA. [7] Department of Pediatrics, University of Rochester School of Medicine and Dentistry, Rochester, NY 14642, USA. [8] Present address: Department of Biotechnology, Jamia Millia Islamia, New Delhi, 110025, India. Correspondence and requests for materials should be addressed to V.B. (email: vineet.bhandari@drexel.edu)

Hyperoxia is a well-known antecedent of injury to developing lungs and is a major contributor to the pathogenesis of bronchopulmonary dysplasia (BPD) in human preterm neonates[1–3]. BPD is the most common chronic lung disease in infants and the long-term consequences extend well into adulthood, with increasing evidence that it can lead to chronic obstructive pulmonary disease (COPD)[4,5]. There is currently no specific preventive or therapeutic agent available to alleviate BPD[6].

MicroRNAs (miRs) are single stranded and evolutionarily conserved sequences of short non-coding RNAs (~21–25 nucleotides)[7] and act as endogenous repressors of gene expression by mRNA degradation and translational repression. They have been shown to have important roles in cell differentiation, development, proliferation, signaling, inflammation, and cell death[7–9]. They have been considered promising candidates for novel targeted therapeutic approaches to lung diseases[7]. Given the role of hyperoxia in development of BPD, a few studies have evaluated expression profiles of miRs in various animal models and human infants[8,10–14].

Angiopoietin-1 (Ang1) is a ligand for receptor tyrosine kinase Tie2[15] expressed on endothelial and epithelial cells[16,17]. Ang1-Tie2 signaling has been shown to be mainly involved in angiogenic activity and promoting maturation of blood vessels, regulated by Akt and MAPK signaling[18–20].

The pulmonary phenotype of BPD is characterized by impaired alveolarization and dysregulated vascularization[21]. Given the potential role of miRs in the pathogenesis of BPD, in this study, we reveal that lung miR-34a levels are significantly increased in neonatal mice lungs exposed to hyperoxia. Deletion/inhibition of miR-34a globally and locally in type 2 alveolar epithelial cells (T2AECs) limits cell death and inflammation with injury and improves the pulmonary and pulmonary arterial hypertension (PAH) phenotypes in BPD mouse models. Conversely, overexpression of miR-34a in room air (RA) worsened the BPD pulmonary and PAH phenotypes, while the addition of miR-34a in the miR-34a deletion mice model exposed to hyperoxia led to reiteration of the BPD pulmonary phenotype. We also show that administration of recombinant Ang1, one of the downstream targets of miR-34a, ameliorates the BPD pulmonary and PAH phenotypes. Finally, using three independent cohorts of human samples, we show the significant association of increased miR-34a and localization to T2AECs in neonates with respiratory distress syndrome (RDS) and BPD. Collectively, our findings support miR-34a as a novel therapeutic target in regulating hyperoxia-induced acute lung injury (HALI) and BPD.

## Results

**Hyperoxia upregulates miR-34a in T2AECs in developing lungs.** miRs have been recently implicated in the regulation of hyperoxia-induced injury and cell death in developing lungs[10,11,22]. Hence, to address the role of miR in hyperoxia-induced lung injury in neonates, we exposed newborn (NB) wild-type (WT) mice to 100% $O_2$ from postnatal day (PN)1–4, and ran a comparative miR array analysis for RA vs. hyperoxia-exposed PN4 mouse lungs (Supplementary Fig. 1A). miR-34a was detected in lungs from WT mice breathing RA and increased markedly after exposure to 100% $O_2$ (Supplementary Fig. 1A). Next, we studied the kinetics of miR-34a expression in hyperoxia-exposed lungs at PN2, PN4, PN7, and PN14 (using lung samples from the mouse model of BPD). miR-34a expression was significantly increased with hyperoxia exposure and reached their maximum levels at PN7 (almost 10-fold); even the BPD model showed a

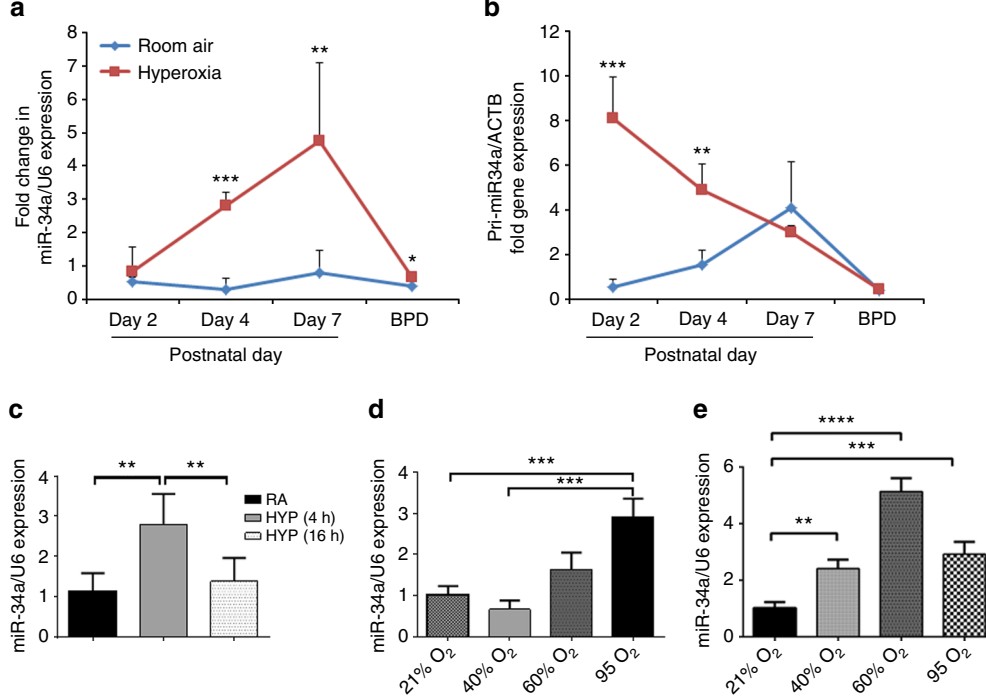

**Fig. 1** Expression of miR-34a in hyperoxia exposed NB lungs and type 2 cells. **a** Representative graphs showing miR-34 expression in WT NB mice exposed to hyperoxia for 2, 4, and 7 days after birth and in the BPD model. **b** Primary miR-34a expression is shown in hyperoxia exposed and BPD murine lung tissue as compared to controls. **c** Freshly isolated type 2 epithelial cells were used for measuring miR-34a expression in room air (RA) and after 4 h and 16 h HYP (95% $O_2$). **d**, **e** MLE12 cells were exposed to different concentrations of oxygen (21, 40, 60, and 95%) for 24 h and 48 h, respectively. NB: newborn; RA: room air; HYP: hyperoxia. A minimum of four animals were used in each group. *$P < 0.05$, **$P < 0.01$, ***$P < 0.001$, ****$P < 0.0001$, compared with controls; 1-way ANOVA

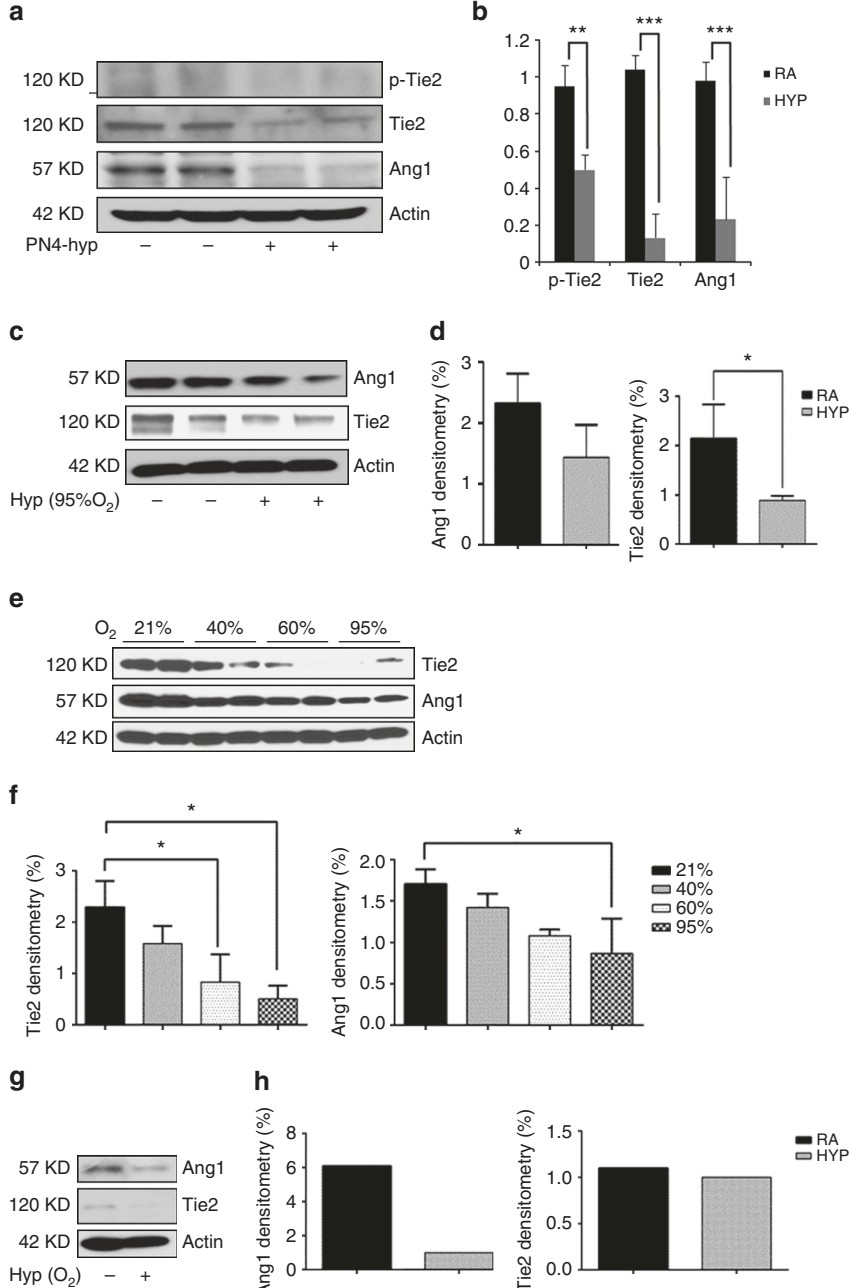

**Fig. 2** Hyperoxia downregulates Ang1-Tie2 signaling in developing lungs. NB WT mice were exposed to hyperoxia from PN day 1–4. **a** Western blots showing decreased expression of Phospho-Tie2, Tie2 and Ang1. **b** Densitometric analysis was completed and the expression of detected proteins was normalized to β-actin. **c, d** MLE12 cells were incubated with or without the presence of hyperoxia (95% $O_2$) for 24 h. Western blot analysis showed downregulation of Tie2 and Ang1, quantified in D. **e, f** Western blot showing decreased Tie2 and Ang1 expression in MLE12 cells, which were exposed to different concentrations of oxygen (21, 40, 60, and 95%) for 48 h, quantified in F. **g, h** Freshly isolated type 2 epithelial cells were incubated in absence and presence of HYP (95% $O_2$) for 4 h and western blot performed for Ang1 and Tie2 expression, quantified in H ($n = 1$). WT: Wild-type; NB: newborn; PN: postnatal; Ang1: Angiopoietin 1; RA: room air; HYP: hyperoxia. Values are means + SEM of a minimum of four observations (in vitro experiments, unless otherwise stated) or four animals (in vivo experiments) in each group. *$P < 0.05$, **$P < 0.01$, ***$P < 0.01$, compared with controls, 1-way ANOVA

significant increase in miR-34a expression as compared to RA control (Fig. 1a; Supplementary Fig. 1B). We next addressed the question whether hyperoxia could increase transcription of miR-34a. Canonically, miRNA genes are transcribed by RNA polymerases into long primary miRNA transcripts (pri-miRNAs). Pri-miRNAs are next cleaved into 60–70 nucleotide-long precursor miRNAs (pre-miRNAs) by the nuclear microprocessor enzymes complex. Pre-miRNAs are next transported to the cytoplasm and

processed to mature form of miRNA[23,24]. In response to hyperoxia, pri-miR-34a was rapidly induced, with the highest expression reaching 15-fold at PN2, after which it began to decline (Fig. 1b), most likely due to the processing of pri-miR-34a into the pre-forms and mature forms. In an effort to localize the specific lung compartment, we checked miR-34a expression in freshly isolated neonatal lung T2AECs, endothelial and macrophage cells. As shown in Fig. 1c, in T2AECs, hyperoxia (95% $O_2$)

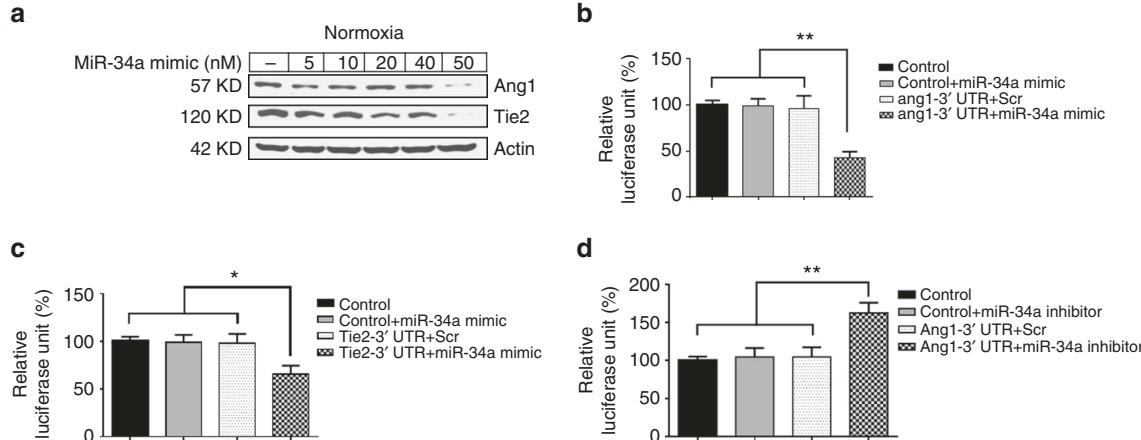

**Fig. 3** miR-34a specifically controls Ang1-Tie2 expression in lung epithelial cells. **a** MLE12 cells were transfected with different concentrations of miR-34a mimic in RA and western blots for Ang1 and Tie2 expression were performed. **b** The wild-type Ang1 3′ UTR reporter vector was co-transfected into the MLE12 cells with either the N.C. mimic or miR-34a mimic **c** The WT Tie2 3′ UTR reporter vector was co-transfected into the MLE12 cells with either the N.C. mimic or miR-34-a mimic. **d** The wild-type Ang1 3′ UTR reporter vector was co-transfected into the MLE12 cells with either the scrambled or miR-34a inhibitor. Ang1: Angiopoietin 1; WT: Wild-type; RA: room air; N.C.: Negative control. Values are means + SEM of a minimum of four observations. *$P$ <0.05, **$P$ <0.01, 2-way ANOVA, Tukey's

gradually induced the expression of mature miR-34a at 4 h. We did not find any significant changes in miR-34a expression in hyperoxia-exposed lung endothelial cells or macrophages (Supplementary Fig. 1C, D). To utilize an in vitro model, we used MLE12 cells, and noted that the expression of miR-34a was highest with 95% $O_2$ exposure at 24 h (Fig. 1d) and with 60% $O_2$ exposure at 48 h (Fig. 1e). Since several publications have shown that miR-34a expression is regulated by Trp53[25,26], we evaluated and noted that Trp53 was acetylated upon hyperoxia exposure to MLE12 cells (Supplementary Fig. 2A). Next, we transfected Trp53 siRNA in MLE12 cells and neonatal PN4 lungs, but only noted a modest (non-significant) decrease in miR-34a expression (Supplementary Fig. 2B, C). We also evaluated miR34a expression in p53 null mutant and Trp53 siRNA treated mice in room air and our BPD model at PN14. These data are shown in Supplementary Fig. 2D, E, where miR34a expression is significantly increased in RA and BPD, compared to WT controls, in p53 absence/inhibition.

Thus, taken together, our data suggest that miR-34a expression is increased upon hyperoxia exposure in developing lungs, and this appears to be localized to T2AECs, of the three lung cell types investigated, as noted above. In addition, miR-34a expression is also regulated by Trp53 in both our in vitro and in vivo hyperoxia-exposed/BPD models.

**miR-34a downregulates Ang1-Tie2 signaling in developing lungs.** To identify the molecular targets of miR-34a, we examined the predicted miR-34a targets using bioinformatics tools, focusing our attention on the regulators of lung inflammation and injury. Using three available prediction algorithms (Targetscan, miR-ANDA, and Pictar), we then produced a comprehensive list of all possible miR-34a targets. We honed onto Ang1 and its receptor, Tie2 (Tek) as potential targets of miR-34a, as they have conserved miR-34a seed sequence in its 3′ UTR (Supplementary Fig. 3A). Ang1 and Tie2 signaling have been consistently demonstrated to be critical players in lung and vascular development[27–29] and several studies have shown Ang1/Tie2 localization to T2AECs[17]. We co-localized Ang1 to T2AECs in neonatal lungs (Supplementary Fig. 3B). These data led us to hypothesize that Ang1/Tie2 may be functional downstream targets of miR-34a in the

inflammatory/apoptotic response to hyperoxia in lung epithelial cells.

The expression levels of Ang1 and Tie2 were first evaluated in hyperoxia-exposed lungs and epithelial cells. As shown in Fig. 2a, b, Ang1 expression was reduced by roughly 70–80% in PN4 hyperoxia-exposed lungs as compared to RA controls. Additionally, levels of Tie2 protein and its phosphorylation were decreased significantly (Fig. 2a, b). Additional downstream targets of miR-34a (Notch2, Sirt1, c-kit, p-ckit, and SCF) were also decreased upon hyperoxia exposure in PN4 neonatal lungs (Supplementary Fig. 3C–E).

We also observed the same effects on Ang1 and Tie2 proteins expression in MLE12 and neonatal mouse primary (freshly isolated) lung T2AECs (Fig. 2c–e). Hyperoxia caused a decrease in Ang1 and Tie2 proteins after 24 h (Fig. 2c, d) and a concentration dependent decrease at 48 h in MLE12 cells (Fig. 2e, f). As in the neonatal lungs, the expression of miR-34a downstream targets were also decreased in MLE12 cells (Supplementary Fig. 3F, G). Interestingly, Trp53 siRNA increased the expression of miR-34a downstream targets Ang1 and Tie2 in MLE12 cells (Supplementary Fig. 3H). In contrast, hyperoxia-exposure to neonatal T2AECs led to decreased Ang1/Tie2 protein levels (Fig. 2g, h) as well as other downstream targets of miR-34a, Sirt1, and Notch2 (Supplementary Fig. 3I).

Next we transfected MLE12 cells with different concentrations of miR-34a mimic and noted that at 50 nm concentration the expression of Ang1 and Tie2 proteins were markedly decreased (Fig. 3a), as well as other downstream targets of miR-34a (Supplementary Fig. 3J). While this observation that miR-34a modulated Ang1/Tie2 signaling in lung epithelial cells suggested an association, it was important to assess whether miR-34a can directly target Ang1/Tie2 according to the in silico target-prediction analysis. To answer this question, we co-transfected miR-34a mimic/inhibitor with Ang1/Tie2 3′ UTRs in MLE12 cells. Overexpression of miR-34a inhibited the activity of a luciferase reporter construct containing Ang1 and Tie2 3′ UTRs (Fig. 3b, c). Similarly miR-34a inhibitor transfection increased 3′ UTR activity of Ang1 only (Fig. 3D). Furthermore, miR-34a inhibitor increased the expression of the downstream targets of miR-34a (Supplementary Fig. 3K).

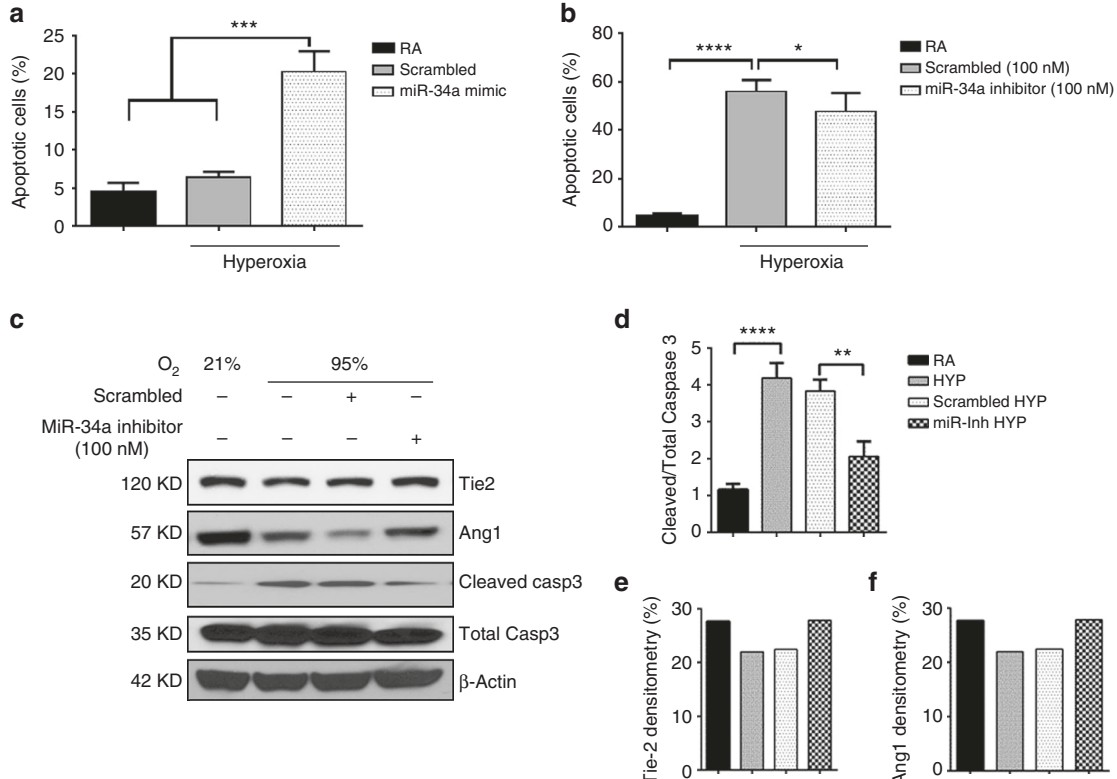

**Fig. 4** miR-34a increases apoptosis in lung epithelial cells. **a** MLE12 cells were transfected with either the N.C. mimic or miR-34a mimic and exposed to hyperoxia for 48 h and subjected to Annexin V and Propidium Iodide (PI) assay. Graph showing significantly increased FACS analysis of Annexin V and PI staining positive cells in miR-34a mimic transfected group compared to its control. **b** Representative graph showing significantly deccreased Annexin V and PI staining positive cells in miR-34a inhibitor transfected group compared to its control in hyperoxia exposed MLE12 cells. **c** Western blot image showing increased Tie2 and Ang1 and decreased cleaved caspase 3 in miR-34a inhibitor transfected group compared to its control in hyperoxia exposed MLE12 cells. **d** Densitometric analysis showing significantly decreased ratio of cleaved caspase 3 to total caspase 3. **e** Densitometric analysis showing increased Tie2 ($n = 1$). **f** Densitometric analysis showing increased Ang1 ($n = 1$). Values are means + SEM of a minimum of four experiments, unless otherwise stated. N.C.: Negative control. $*P < 0.05$, $**P < 0.01$, $***P < 0.001$, $****P < 0.0001$ compared with controls, 1-way ANOVA

**miR-34a increases apoptosis in lung epithelial cells**. We next evaluated the role of miR-34a in the regulation of hyperoxia-induced cell death in MLE12 cells. We transfected these cells with miR-34a inhibitor, miR-34a mimic and scrambled controls and exposed to 48 h hyperoxia. Cells cultured in 5% $CO_2$ and RA did not show significant cell death (Supplementary Fig. 4, Fig. 4a). In contrast, 95% $O_2$ (with scrambled controls) caused a modest increase in cell death, mostly in Annexin V positive staining after 48 h in hyperoxia (Supplementary Fig. 4, Fig. 4a). This hyperoxia-induced cell death response was significantly increased in the presence of miR-34a mimic (mostly Annexin V+Propidium iodide positive) and decreased with miR-34a inhibitor (mostly Annexin V positive) transfection as compared to scrambled controls (Supplementary Fig. 4, Fig. 4a, b). In addition, miR-34a-inhibitor treatment increased miR-34a targets Ang1 and Tie2 expression in hyperoxia-exposed MLE12 cells as compared to scrambled control (Fig. 4c). Furthermore, cleaved caspase3 expression was also decreased in miR-34a-inhibitor treated group (Fig. 4c, d). Quantification of Tie2 and Ang1 is shown in Fig. 4e, f.

Taken together, these studies show that miR-34a stimulates epithelial cell death and decreases the expression of target proteins Ang1/Tie2; conversely, miR34a-inhibition decreases cell death and enhances the expression of target proteins Ang1/Tie2 upon hyperoxia exposure in vitro.

**miR-34a global deletion renders mice resistant to HALI/BPD.** To determine the contribution of miR-34a to HALI, we examined WT and miR-34a (−/−) mice exposed to hyperoxia and noted that miR-34a (−/−) NB mice in hyperoxia had better survival than WT mice (Fig. 5a). In the HALI model, as compared to NB WT, hyperoxia exposure to NB miR-34a (−/−) mice lungs at PN7 had improved lung morphometry, as demonstrated by chord length measurements (Fig. 5b, c). The same trend was observed in inflammatory bronchoalveolar lavage fluid (BALF) interleukin (IL)-1β and IL-6 levels, both of which were increased in WT, but not as much in miR-34a (−/−) mice, with IL-6 levels being significantly decreased, at PN7, upon hyperoxia-exposure (Supplementary Figs. 5A, 5B).

We next determined the inflammatory markers, specifically neutrophil influx, and myeloperoxidase (MPO) activity in the BPD model at PN14. Hyperoxia-exposed WT mice had a maximal increase in neutrophils and MPO activity, which were significantly decreased in the miR-34a (−/−) mice lungs (Fig. 5d, e).

In the BPD model, as compared to NB WT, NB miR-34a (−/−) mice BPD lungs had improved lung morphometry, as demonstrated by chord length measurements (Fig. 5f, g) which correlated with TUNEL staining (Fig. 5h). In addition, there was significantly increased Ang1, Tie2, SCF, and Notch2 expression in the miR34a (−/−) BPD mice lungs (Fig. 5i). In contrast, we noted decreased expression of Ang2 in miR34a (−/−) mice lungs upon hyperoxia exposure at PN4 as well as being significantly decreased in BPD mice lungs at PN14, compared to respective controls (Fig. 5j, k).

Collectively, these data demonstrate that miR-34a is a critical component of the neonatal mouse response to hyperoxia and regulates inflammation and alveolarization in HALI and BPD.

**Deletion of T2AEC-specific miR-34a reverses the BPD phenotype.** To address the role of miR-34a being specifically expressed in T2AECs in mediating alveolarization, we used *SPC-CreER/miR-34a^{fl/fl}* mouse line to disrupt *miR34a* specifically in

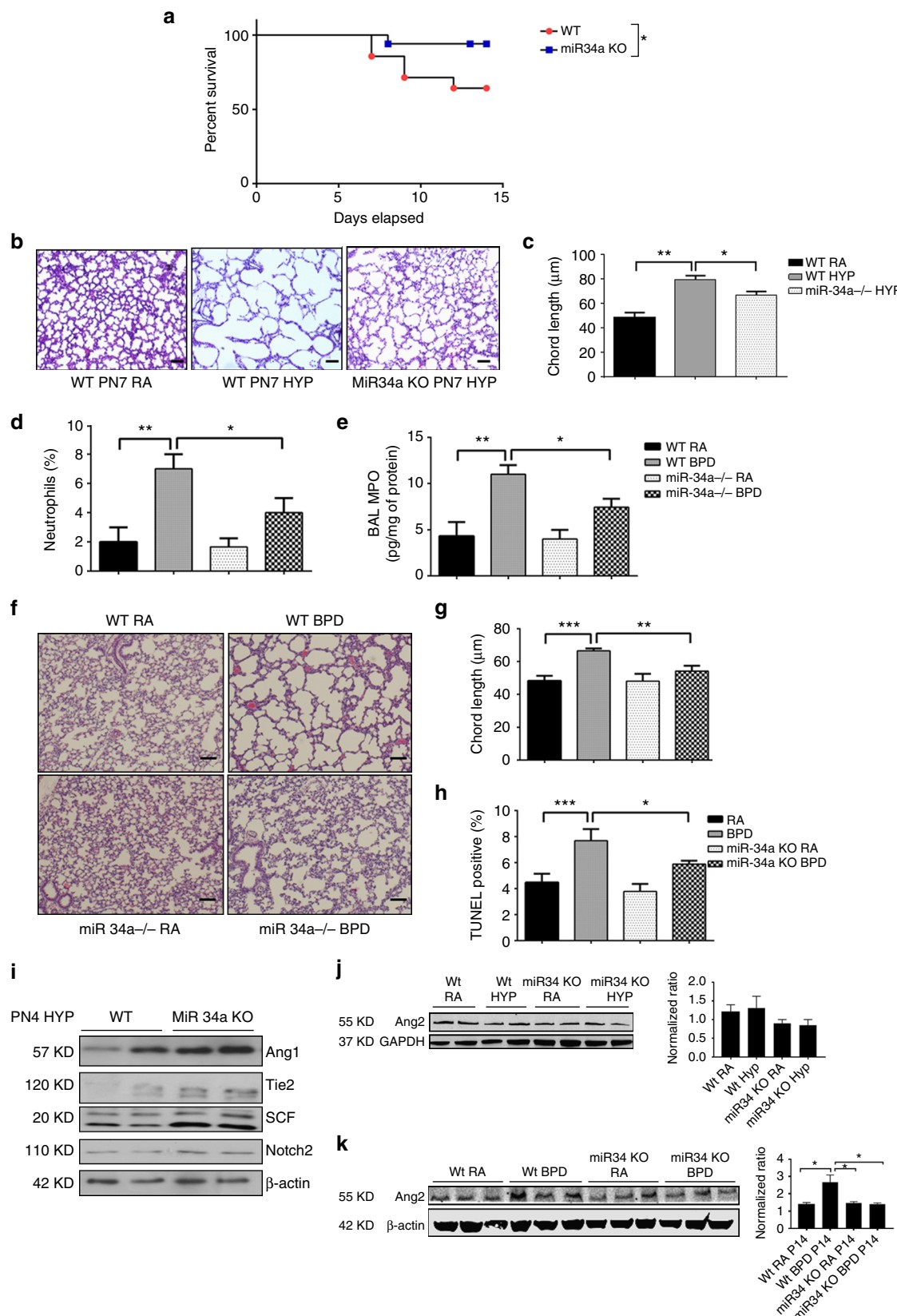

T2AECs. Expression of Cre recombinase is activated in T2AECs by the tamoxifen-inducible system coupled with T2AEC-specific promoter SpC. To determine whether the increased T2AEC expression of miR-34a seen in the above studies is causally related to impairment of lung BPD phenotype, we used mice in which miR-34a deletion was conditionally induced exclusively in T2AECs. We crossed miR-34a-floxed mice (miR-34a^{fl/fl}) with SPC-Cre-ER mice. SPC-Cre-ER mice express Cre recombinase in SP-C positive T2AECs in a tamoxifen-inducible manner. miR-34a deletion was induced by maternal tamoxifen administration (2 mg) from PN1–PN5 in the NB pups, via maternal milk. Tamoxifen-induced Cre recombinase activity markedly decreased miR-34a expression in PN4 lung (Fig. 6a). We confirmed that tamoxifen injection alone had effect on lung morphometry (Supplementary Fig. 5C) in WT, Cre, or miR-34a^{fl/fl} mice.

Importantly, we observed improved chord lengths in T2-miR34a^{−/−} BPD lungs as compared to appropriate controls (Fig. 6b). Additionally, T2AECs miR-34a deletion decreased TUNEL-positive cells (Fig. 6c) and lung inflammation as demonstrated by a decrease in lung neutrophils in the BALF and a significant decrease in tissue MPO activity in T2-miR34a^{−/−} lungs (Fig. 6d, e).

Hence, miR-34a deletion in T2AECs is sufficient to protect the newborn lung to develop the BPD pulmonary phenotype, upon hyperoxia-exposure.

**miR-34 overexpression in RA restores the BPD phenotype**. To address whether miR-34 expression was required and sufficient for the hyperoxia-induced lung injury and inflammation leading to the BPD pulmonary phenotype, we next asked whether only miR-34a overexpression itself was sufficient, in the absence of hyperoxia i.e., in RA. To test this, we intranasally administered miR-34a mimic in WT and miR-34a (−/−) mice, and confirmed the restoration of miR-34a levels (Supplementary Figs. 6A, 6B). Figures 7a, b show that administration of miR-34a mimic is sufficient to elicit the BPD phenotype in RA.

Furthermore, restoring miR-34a levels by intranasal administration of miR-34a mimic in miR34a (−/−) animals recapitulated the BPD phenotype induced by hyperoxia (Fig. 7c). Mechanistically, we were able to show that miR-34a mimic in RA was able to decrease the expression of the downstream targets (Ang1, Tie2, SCF, c-kit, Notch2, and Sirt1) in MLE12 cells as well as in vivo (Fig. 7d, e).

Taken together, our data show that T2AEC-specific deletion of miR-34a is sufficient to rescue the BPD phenotype in hyperoxia; conversely, increased expression miR-34a in RA is sufficient to re-create the BPD pulmonary phenotype. In addition, provision of miR-34a to the miR34a (−/−) BPD model re-creates the BPD pulmonary phenotype. These effects are associated with differential regulation of downstream targets of miR-34a, impacting on inflammatory and angiogenic pathways.

**miR-34a inhibitor treatment improves hyperoxia-induced BPD**. Given that genetic deletion of miR-34a was associated with complete protection from hyperoxia-induced changes in lung morphometry and inflammation, we next sought to block miR-34a as a therapeutic strategy in NB WT mice exposed to hyperoxia, using a miR-34a inhibitor via the intranasal route. We administered 5 µl (20 µM concentration) of miR-34a inhibitor (or scrambled control) at PN2 and PN4 intranasally, during hyperoxia exposure. Lung histology showed that, in comparison to the scrambled group, intranasal treatment with miRNA-34a inhibitor in neonatal mice significantly improved the BPD pulmonary phenotype, specifically in terms of chord length and septal thickness (Fig. 8a–c). Administration of miR-34a inhibitor in BPD mice also reduced the TUNEL-positive score (Fig. 8d) and reduced cleaved-caspase3 expression (Fig. 8e). MiR-34a treatment was also effective in reducing lung inflammation as evident by reduced neutrophil infiltration, MPO activity, IL-1β and IL-6 in BALF (Fig. 8f–i). To examine lung regenerative capacity in PN4 mouse lungs, these were stained with PCNA, which revealed increased cell proliferation in the lungs treated with miR-34a inhibitor (Fig. 8j). Finally, in comparison to controls, miR-34a inhibitor treated mice had significantly increased Ang1 and Tie2 as well as Sirt1 and Bcl2 protein expression in hyperoxia-exposed lungs (Fig. 8k).

**miR-34a inhibition improved PAH in the mouse BPD model**. In addition to impaired alveolarization, dysregulated vascularization is a key component of the pathology of BPD lungs. Hence, to understand the impact on vascular development, we investigated the effects of neonatal hyperoxia on the vessel density in these animals by immunostaining the small non-muscularized vessels with Willebrand Factor (vWF)—a marker for endothelial cells. As previously reported[30], we observed decreased vascular growth in BPD animals compared to RA mice lungs, which was improved in miR-34a inhibitor treated animals, confirmed by quantification (Supplementary Fig. 6C, D). Importantly, as was the case with alveolarization (Fig. 7a–c), there was decreased vascular density (equivalent to control and scrambled miR-treated BPD lungs) in the miR34a-mimic treated miR-34a (−/−) mice hyperoxia-exposed BPD lungs (Supplementary Fig. 6C, D).

Another critical element of BPD is the associated PAH, as noted in the mouse model[31,32] and human BPD[33]. miR-34a inhibitor treated animals demonstrated attenuated right ventricular hypertrophy (RVH), as indicated by right ventricle (RV)/left ventricle (LV) ratio and Fulton's Index (Supplementary Fig. 6E, F). Importantly, the PAH indices worsened upon exposure to

**Fig. 5** Deletion of miR-34a results in improvement of BPD. **a** NB WT ($n = 8$) and miR-34a KO ($n = 11$) mice were exposed to hyperoxia from PN day 1–15 and were monitored for survival. Survival data were analyzed using the Kaplan−Meier method and log-rank test. **b** Representative images of lung histology (H&E stain) of NB miR-34a KO mice exposed to RA or 100% O$_2$ at PN7. Scale bar: 100 µm. **c** Bar graph showing the morphometric analysis of lung histology sections of NB miR-34a KO mice exposed to RA or 100% O$_2$ at PN7. **d, e** Hyperoxia increased the numbers of neutrophils and BAL myeloperoxidase (MPO) in neonatal mouse lungs, and the deletion of miR-34a attenuated the hyperoxia-induced increase in neutrophil numbers in the BPD mouse model. **f** Representative H&E stained images of alveolar regions from lungs from WT and miR-34a KO mice from RA and BPD groups. **g** Morphometric analysis of lung histology sections of NB WT and miR-34a KO expressed as chord length and analyzed using Image J software. **h** Bar graph showing the percentage of TUNEL-positive cells indicating the apoptosis quantification in WT and miR-34a KO BPD models. **i** NB WT and miR-34a KO mice were exposed to hyperoxia from PN day 1–4. Western blots showing increased expression of Tie2, Ang1, SCF, and Notch2 in miR-34a KO lungs as compared to WT. **j** NB WT and miR-34a KO mice were exposed to hyperoxia from PN day 1–4. Western blots and quantification showing decreased expression of Ang2 in miR-34a KO lungs compared to WT, upon exposure to hyperoxia ($n = 2$). **k** Western blots and quantification of the same showing significantly decreased expression of Ang2 in miR-34a KO lungs as compared to WT, in the BPD model at PN14 ($n = 3$). BPD: bronchopulmonary dysplasia; NB: newborn; WT: Wild-type; KO: knockout or null mutant; PN or P: postnatal; BAL: bronchoalveolar lavage. Values are means ± SEM of a minimum of four animals in each group, unless otherwise stated. *$P < 0.05$, **$P < 0.01$, ***$P < 0.001$, 2-way ANOVA, Tukey's

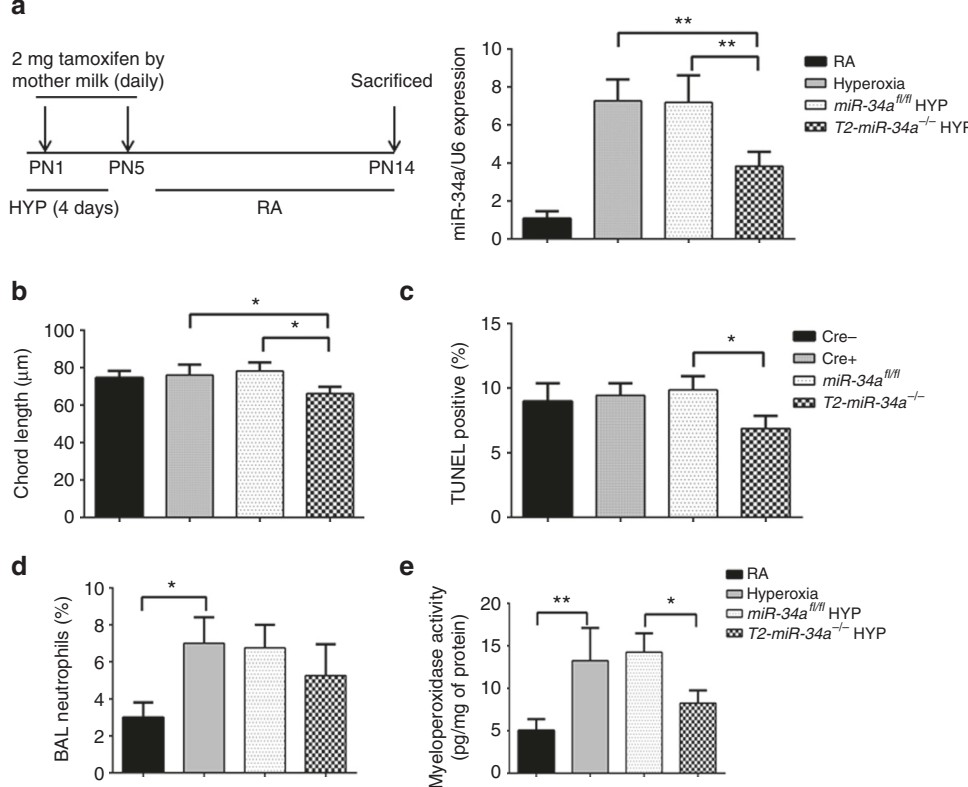

**Fig. 6** Inducible deletion of miR-34a from type 2 epithelial cells improves BPD. **a** Schematic of tamoxifen-induced deletion of miR-34a in Spc CRE-expressing *miR-34a^fl/fl* mice. From the birth of NB pups, dams were injected with 2 mg tamoxifen IP for four consecutive days (PN1-PN5). For the BPD model pups were kept in the hyperoxia chamber for 4 days and both dams (alternating in RA and hyperoxia exposure every 24 h) were given tamoxifen. Representative bar graph showing tamoxifen deletion of miR-34a in Spc CRE positive *miR-34 KO* lungs (T2-miR34a^−/−). **b** Representative graph shows chord length analysis of lung histology (H&E stain) of NB *T2-miR34a^−/−* mice BPD model along with controls. **c** Bar graphs showing less TUNEL count of lung histology sections of NB *T2-miR34a^−/−* mice BPD model, as compared to controls. **d, e** BAL neutrophils count and myeloperoxidase activity were reduced in NB *T2-miR34a^−/−* mice BPD model, as compared to controls. BPD: bronchopulmonary dysplasia; Spc: surfactant protein c; NB: newborn; IP: intraperitoneal; PN: postnatal; RA; room air; KO: knockout or null mutant; T2: type 2 alveolar epithelial cells. Values are means ± SEM of a minimum four animals in each group. *$P < 0.05$, **$P < 0.01$, 2-way ANOVA, Tukey's

miR34a-mimic in the miR34a-mimic treated *miR-34a* (−/−) hyperoxia-exposed mice (Supplementary Fig. 6E, F). Similar protective responses were noted in the *miR-34a* (−/−) mice (Supplementary Fig. 6G, H). Using micro-CT, we were able to confirm that the improved PAH indices were secondary to improved numbers of large and medium-sized pulmonary vessels (Supplementary Fig. 6I, J).

**Ang1 is protective of the BPD pulmonary and PAH phenotypes**. To firmly establish the mechanistic role of miR-34a, given the impact on Ang1 expression, we evaluated the effect of Ang1 administration in the regulation of lung epithelial cell survival pathway and the BPD pulmonary phenotype. In the NB WT murine model of BPD, administration of recombinant Ang1 resulted in a significant decrease in TUNEL-positive cells at PN14 (Fig. 9a). Concomitantly, we noted a significant improvement in alveolarization, as evidenced by decreased chord length (Fig. 9b). In addition, in the PN7 HALI model, Ang1 treatment showed improved Ki67 staining levels similar to that of the *miR-34* (−/−) mice lungs (Supplementary Fig. 7).

Furthermore, Ang1 treatment was protective of BPD-associated PAH (Fig. 9c, d). Since Akt and Erk pathways are important for epithelial cell survival and growth in response to hyperoxia[34] and Ang1/Tie2 signaling activates Akt and Erk by phosphorylating them, we examined the effect of miR-34a

inhibitor in vitro. Phosphorylation of Akt and Erk was increased in miR-34a inhibitor transfected cells treated with recombinant Ang1 (Fig. 9e).

Taken together, our data suggest that miR-34a inhibitor treatment improves the alveolar and vascular development in the hyperoxia-exposed BPD mouse model, at least in part, via the Ang1/Tie signaling pathway.

**miR-34a regulates epithelial-mesenchymal transition in BPD**. Investigators have reported a role for miR34a in TGF-β1 and drug-induced epithelial-mesenchymal transition (EMT) in alveolar type II cells[35]. We evaluated 2 EMT markers (N-cadherin and E-cadherin) in lung homogenates of WT and miR-34a null mutant mice upon hyperoxia exposure at PN4 and in the BPD model at PN14 (Supplemental Fig. 8). We noted that the mesenchymal marker (N-cadherin) is decreased in the miR-34a null mutant mice, as compared to WT, in room air, PN4 hyperoxia and BPD lungs at PN14, with no change in the epithelial marker (E-cadherin). Overall, our results would suggest that there is a potential role for miR-34a regulating EMT in the BPD model, but further experiments (beyond the scope of this manuscript) would be required to be definitive.

**miR-34a regulation in other models BPD/lung injury**. We[36,37] and others[38] have reported that overexpression of transforming

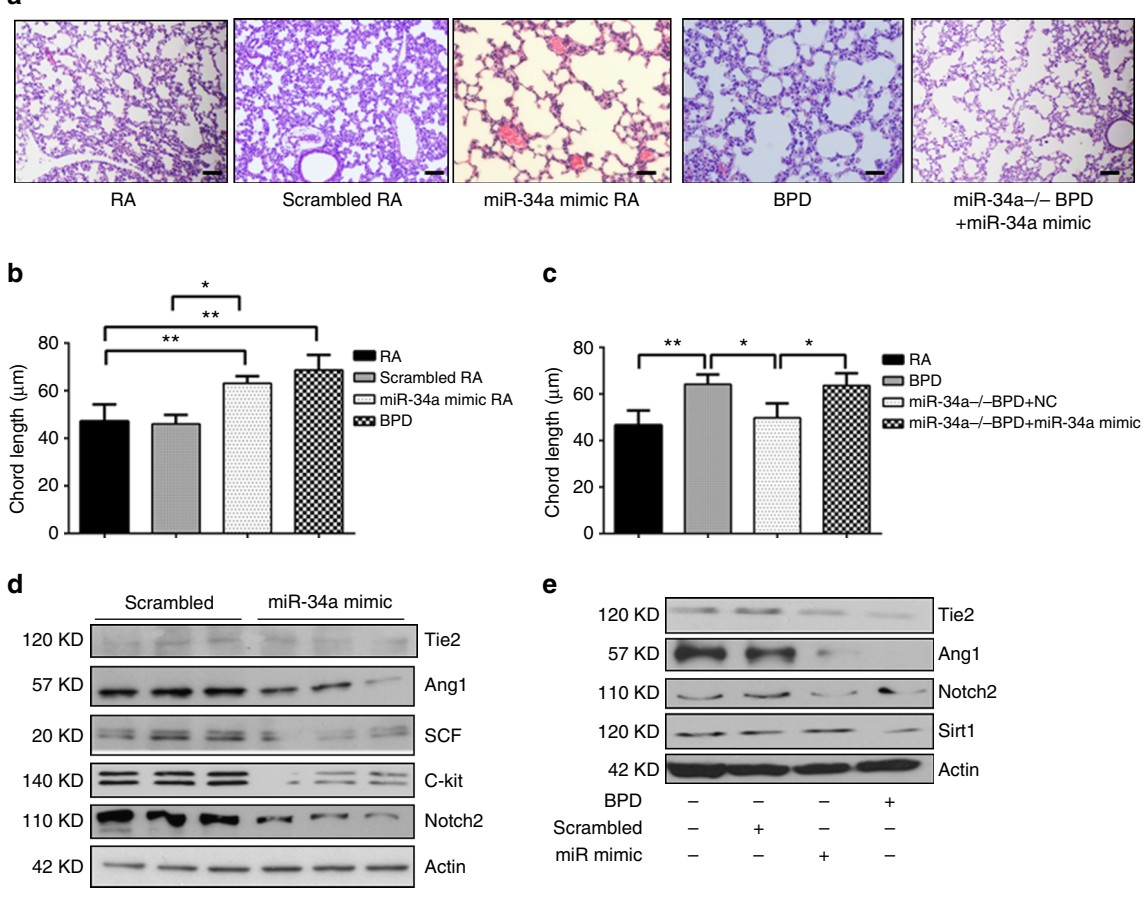

**Fig. 7** miR-34a overexpression in lungs results in the BPD phenotype. **a** Representative images of lung histology (H&E stain) of NB WT mice models of RA or BPD were treated with miR-34a mimic (20 μM; PN2 and PN4) intranasal. Scale bar: 100 μm. **b** Morphometric analyses of lung histology sections of NB WT mice RA or BPD at PN14. Alveolar size expressed as chord length analyzed by Image J software. **c** Morphometric analyses of lung histology sections of NB WT mice RA or BPD treated with miR-34a mimic and control. **d** Western blot analysis of Ang1, Tie2, SCF, c-Kit, and Notch2 was performed on MLE12 cells transfected with miR-34a mimic or scrambled control **e** Western blotting of Tie2, Ang1, Notch2, and Sirt1 was performed on miR-34a mimic treated RA and BPD mice lungs. BPD: bronchopulmonary dysplasia; NB: newborn; PN: postnatal; WT: wild-type; RA: room air; Ang1: angiopoietin 1; SCF: stem cell factor. Values are means ± SEM of a minimum of four animals (in vivo experiments) in each group. *$P < 0.05$, **$P < 0.01$, compared with controls, 1-way ANOVA

growth factor (TGF)-β1 in the developing lung mimics the BPD pulmonary phenotype. To determine whether the effects of miR-34a are limited to the hyperoxia-induced BPD model or could be dependent on other injury mediators, we tested the expression of miR-34a in lungs of our TGF-β1 doxycycline-inducible overexpressing transgenic mouse model. We noted increased expression of miR-34a at PN10 in the neonatal mouse lungs as compared to transgene negative animals (Supplementary Fig. 9A). Furthermore, we also found miR-34a targets Ang1 and Sirt1 were decreased in TGF-β1 TG mouse lung samples (Supplementary Fig. 9B). Trp53 (p53) expression was also increased in TGF-β1 transgenic neonatal mouse lung tissues (Supplemental Fig. 9B).

We further investigated the role of hypoxia and TGFβ signaling in a newborn mice model[39]. We obtained lung tissues and noted that miR34a expression was decreased with hypoxia and decreased TGFβ signaling (using inducible dominant-negative mutation of the TGF-beta type II receptor (DNTGFbetaRII) mice) or a combination of the two exposures (Supplementary Fig. 9C). Furthermore, we used antenatal LPS administration (mimicking chorioamnionitis) with/without additional PN hyperoxia exposure in a neonatal rat model, and noted increased expression of miR34a in the lungs, but only when PN hyperoxia exposure was present (Supplemental Fig. 9D).

Taken together, our data would suggest that hyperoxia as well as increased TGFβ signaling is a major contributor to the increased levels of miR34a.

**Human BPD infant lungs have increased miR-34a expression.** To evaluate the human disease relevance of these findings, we examined whether miR-34a is increased in the TA and/or lungs of babies with RDS/BPD. The expression of miR-34a was significantly higher in TA cell pellets from individuals who went on to develop BPD and/or died, compared to controls (Fig. 10a). Similarly, in situ hybridization showed higher expression of miR-34a in epithelial linings of lungs of neonates with RDS especially with RDS 3-7 and RDS >7 days of PN age, mostly localized to T2AECs (Fig. 10b).

For further validation, we used a third independent collection of human lung samples matched by gestational and/or PN age,[40] and conducted an immunoblot analysis. As noted in Fig. 10c–e, there was a marked decrease in the Ang1 and Tie2 proteins, comparing term control infants with no lung disease to those with evolving and established BPD[1]. Upon densitometry quantification, it was obvious that the Ang1/Tie2 proteins were decreased with increasing severity of disease, with the maximal decrease noted in those with established BPD (Fig. 10c–e).

These data would suggest that hyperoxia and/or ventilation-induced injuries to the developing lung are accompanied by alterations in the miR-34a-Ang1/Tie2 signaling axis in human neonates. A proposed schema for the role of miR-34a in the pathogenesis of BPD is shown in Fig. 10f.

## Discussion

The present study reports on three major novel findings. First, hyperoxia induces miR-34a expression in lung T2AECs of newborn mice and human infants with the clinically relevant diagnoses of RDS, evolving and established BPD suggesting

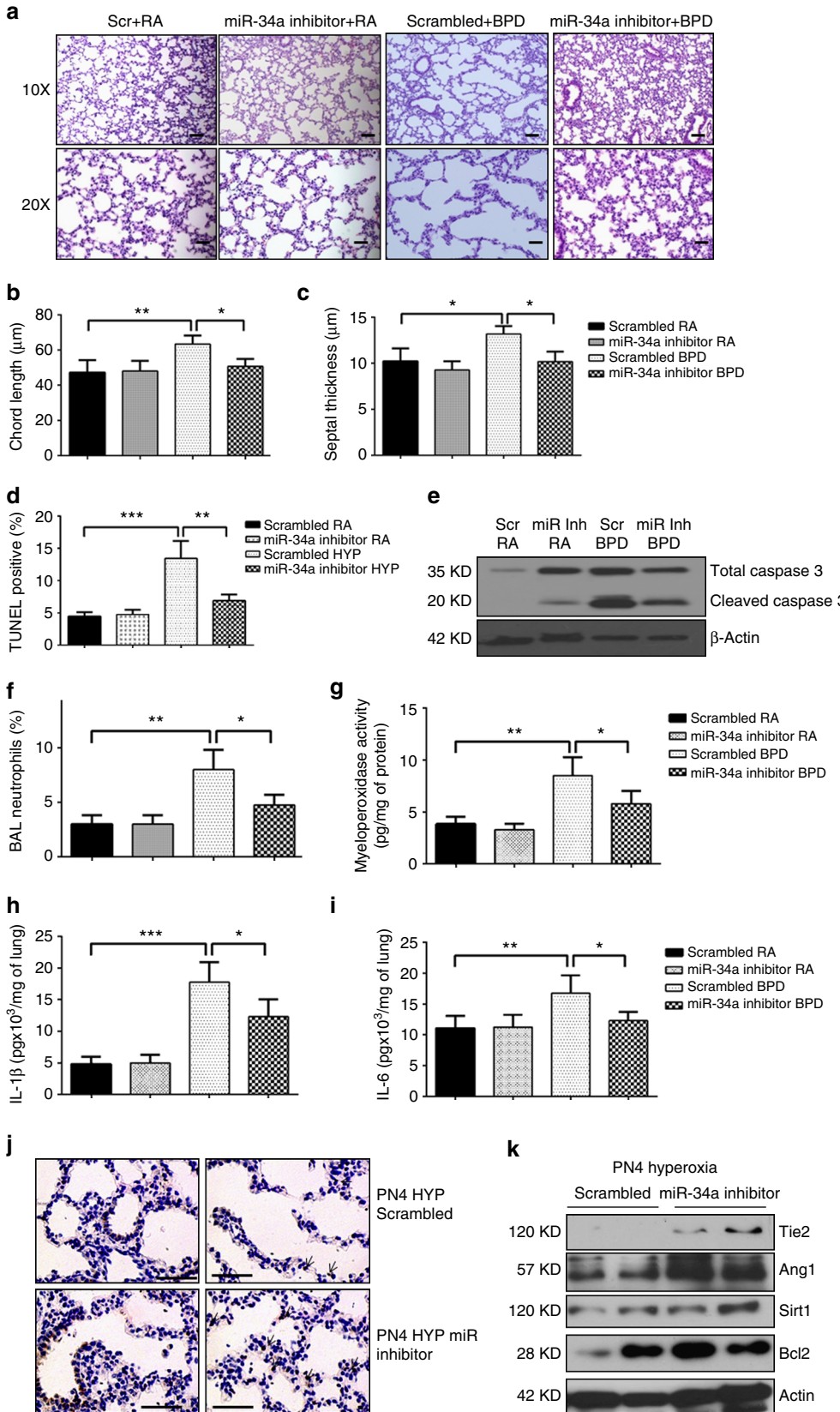

translational significance. Second, using genetic gain-of-function and loss-of-function strategies (including deletion of miR-34a specifically in T2AECs), we comprehensively prove a causal detrimental role of increased miR-34a; conversely, inhibition of miR-34a was protective of the BPD pulmonary and associated PAH phenotypes. Third, we experimentally validate the mechanistic angiogenic, inflammatory, cell death and cell proliferation pathways of miR-34a, focusing on the role of vascular downstream targets, Ang1 and Tie2, and show that Ang1 treatment is protective of the BPD pulmonary and associated PAH phenotypes.

The role of miRNAs in epithelial cells related to inflammatory and immune responses has been demonstrated by various groups[22,41–43]. Recently Narasaraju et al.[22] showed decreased miR-150 expression in alveolar epithelium in neonatal mouse upon hyperoxia exposure, which might be responsible for epithelial apoptosis. Similarly overexpression of miR-181b resulted in the induction of an increment in IL-6 levels in bronchial epithelial cells[43]. The miR-200 family was significantly upregulated during T2AECs differentiation in fetal lung; miR-200 induction was inversely correlated with expression of known targets, transcription factors ZEB1/2, and TGF-β2. miR-200 antagonists inhibited thyroid transcription factor (TTF)-1 and surfactant proteins and upregulated TGF-β2 and ZEB1 expression in T2AECs[44].

Several studies have recently examined the role of specific miRNAs in the pathogenesis of lung injury. Accumulating studies have implicated a role of miRNAs in lung diseases such as adult RDS (ARDS), fibrosis, COPD, and BPD[45–48]. miR-206 was reduced in BPD mice compared with controls and in BPD patients compared with controls. MiR-206 overexpression significantly induced cell apoptosis, reduced cell proliferation, migration, and adhesion abilities, whereas the inhibition of miR-206 expression had the opposite effect[12]. Recently, decreased miR-489 has been reported upon hyperoxia exposure in neonatal mice and humans with BPD[49]. The authors suggest that decreased miR-489 may be inadequate attempts at compensation[49]. Another group has reported that miR-17~92 expression is significantly lower in human BPD lungs[50]. While previous studies have reported the expression of miR-34a in neonatal and adult lung injury[11,51], none, to the best of our knowledge, has comprehensively mechanistically defined the role of miR-34a in HALI and BPD in developing lungs.

We provide evidence of the in vivo relevance of miR-34a in hyperoxia-induced neonatal human and murine lung injury. Moreover, we identify the underlying molecular mechanisms by analyzing specific inflammatory/vascular/survival-associated targets of miR-34a. Most importantly, we demonstrate the feasibility and efficacy of in vivo miR-34a inhibition as a protective therapeutic option to ameliorate BPD and associated PAH. Recent direct evidence suggests that miR-34a is correlated with potential inflamed states, including the staphylococcal enterotoxin B-induced acute inflammatory lung injury[51], hepatic ischemia/reperfusion injury[52], high-fat diet induced hepatic steatosis[53],

cardiac aging, and myocardial infarction[54–56] and acute kidney injury[57]. Therefore, miR-34a may be an indicator of inflammation/injury, especially since its role in cell death and cell cycle is well established[58–60]. Other studies have indicated that miR-34a attenuates cell proliferation, invasion and EMT[35,61,62]. Consistent with above studies, we also observed that when miR-34a was augmented in neonatal lung, cell proliferation, and angiogenesis levels were notably attenuated, and apoptosis was significantly increased. Furthermore, downregulated miR-34a had the opposite effect suggesting that miR-34a can significantly affect cellular biological function.

To identify the downstream mechanism of miR-34a-regulated protection, we used mRNA databases to identify targets and revealed many apoptosis/inflammation associated genes. We focused on Ang1, Tie2, Sirt1, Notch2, and Bcl2, all of which have established roles in recovery of lung injury and/or apoptosis. Ang1 and Tie2 are protective as regards DNA-damage and oxidative stress and were among the strongest downregulated targets in our profiling approach. Sirt1, Notch2, CDKs, and Bcl2 are predicted targets of miR-34a and these have previously been described as important factors in lung development and injury[63–66]. We have previously reported lower levels of Sirt1 to be associated with BPD in human neonates[67]. Notch2 expression has been reported to be decreased upon hyperoxia exposure to newborn rats[68]. Interestingly, reconstitution of rAng1 in miR-34a overexpressing epithelial cells underlined their critical importance in miR-34a-mediated effects on cell survival regulation. In vivo, Ang1/Tie2 was significantly upregulated by miR-34a antagonism, and this signaling was able to ameliorate the neonatal BPD phenotype. Ang1 secretion has been shown to be responsible for restoring epithelial protein permeability through suppression of NFκB activity in human T2AECs[69]. Combined VEGF and Ang1 gene transfer has been reported to mature the new vasculature, reducing the vascular leakage seen in VEGF-induced capillaries[70]. Our data thus reveal a critical role of miR-34a and the downstream Ang1/Tie2 signaling and the transition between the pro-inflammatory and anti-inflammatory phenotypes, which is believed to be important for the molecular regulation of functional shaping of T2AECs apoptosis and proliferation and the related BPD phenotype. We thus propose Ang1/Tie2 signaling as the major factor in miR-34a-mediated BPD.

In support, using complementary gain-of-function and loss-of-function approaches, we demonstrated that miR-34a inhibits downstream cell survival signaling by directly targeting Ang1/Tie2 in vitro and in vivo, an effect that inhibits apoptosis, neutrophil accumulation, and vascular injury in vivo. Importantly, lung delivery of miR-34a inhibitor repressed activation of the cell death pathway and reduced the lung BPD phenotype in hyperoxia-exposed mice. Moreover, miR-34a overexpression in room air alone was sufficient to produce the BPD phenotype in neonatal mice (Fig. 7a, b).

We identified patients with RDS, evolving and established BPD as having high levels of miR-34a in lung and TA cell pellets

**Fig. 8** miR-34a inhibition improves BPD phenotype via increased Ang1-Tie2 signaling. **a** Representative images of lung histology (H&E stain) of NB WT mice models of RA or BPD were treated with miR-34a inhibitor (20 μM; PN2 and PN4) intranasal. Scale bar: 100 μm. **b, c** Morphometric analyses of lung histology sections of NB WT mice miR-34a inhibitor at PN14. Alveolar size expressed as chord length and septal thickness. **d** Representative graph shows TUNEL-positive cells (%) in NB WT mice lungs treated with miR-34a inhibitor or control. **e** Western blot analysis of cleaved and total Caspase 3 was performed on MLE12 cells transfected with miR-34a inhibitor or scrambled control. **f, g** Bar graphs showing BAL neutrophils count and myeloperoxidase activity in RA and BPD mice treated with miR-34a inhibitor or scrambled control. **h, i** Bar graphs showing lung IL-1β and IL-6 in RA and BPD mice treated with miR-34a inhibitor or scrambled control. **j** Representative immunohistochemistry (IHC) images showing increased staining of PCNA in miR inhibitor treated PN4 hyperoxia exposed animals. Scale bar: 100 μm. **k** Western blot shows increased Ang1 and Tie2 in miR-34a inhibitor treated PN4 lung samples. BPD: bronchopulmonary dysplasia; Ang1: angiopoietin 1; NB: newborn; WT: wild-type; RA: room air; PN: postnatal; IL: interleukin. Values are means + SEM of a minimum of four observations (in vitro experiments) or four animals (in vivo experiments) in each group. *P <0.05, **P <0.01, ***P <0.01, compared with controls, 1-way ANOVA

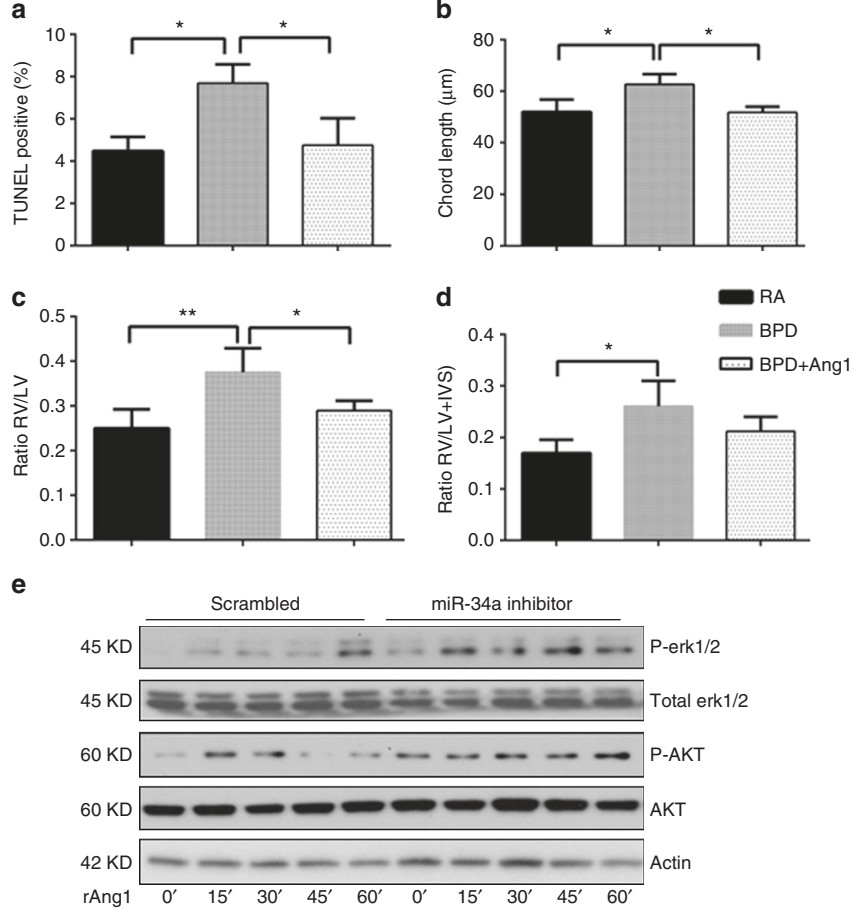

**Fig. 9** Improved effects of miR-34a inhibition occurs via Ang1-Tie2 signaling. **a**, **b** Representative bar graphs show improved TUNEL and morphometry in recombinant Ang1 (rAng1) treated mouse lungs. **c**, **d** Improved RV to LV and RV/(LV + IVS) ratios were noted in rAng1 treated BPD mice as compared to controls. **e** Activation of Erk and Akt signalling by rAng1 in miR-34a inhibitor or scrambled control transfected MLE12 cells as analyzed by western blot of phosho-Erk1/2, total Erk1/2 protein, phospho-Akt and total Akt protein, shown at the indicated time points. Ang1: angiopoietin 1; RV: right ventricle; LV: left ventricle; IVS: interventricular septum. Values are means ± SEM of a minimum of four observations (in vitro experiments) or four animals (in vivo experiments) in each group. *$P < 0.05$, **$P < 0.01$, compared with controls, 1-way ANOVA

compared with "No BPD" term control patients. Ang1/Tie2 signaling was decreased in RDS, evolving and established BPD patient lungs as compared to controls (Fig. 10a–e). These findings coupled with the improved survival observed in miR-34a "knockout" studies in neonatal BPD mice (Figs. 5, 6, 8) suggest that therapies directed at inhibiting miR-34a expression may ameliorate BPD.

Of note, miR-34 family members also have been recognized as tumor suppressor miRNAs. Given that the miR-34 family has been implicated in the p53 tumor suppressor network, and that p53 pathway defects are common features of human cancer[25], miR-34 inhibition therapy is considered a promising therapeutic approach[26]. Recent reports demonstrate that inhibition of the miR-34 family does not promote tumorigenesis, supporting the potential for therapeutic suppression of this family as a treatment for BPD[56]. It is important to emphasize that future therapeutic approaches using a targeted approach that can restrict inhibition to the lung T2AECs may have the maximal therapeutic potential. Such an approach can be done by utilizing surfactant (which is used routinely in preterm neonates with RDS) as a delivery vehicle for a miR-34a inhibitor. This will not only rapidly and effectively deliver the drug to the alveolar compartment, but will also allow uptake by T2AECs, as surfactant is recycled within the lung.

Another recently described complication of BPD has been BPD-associated PAH[33]. We found that our BPD model also led to PAH, which was associated with decreased vascular development

(Supplementary Fig. 6C–H). Importantly, it should be pointed out that miR-34a deletion/inhibition enhanced pulmonary vascular development and indices of PAH in hyperoxia-exposed neonatal mice. These data suggest that the mechanism of PAH in BPD mice is, to some degree, regulated by the miR-34a/Ang1/Tie2 axis.

In conclusion, we show that miR-34a contributes to neonatal murine BPD by influencing T2AECs apoptosis through regulation of anti-apoptotic *Ang1/Tie2* signaling. Silencing of miR-34a ameliorates the apoptotic response in vitro and in vivo, leading to suppressed epithelial apoptosis; this, in turn, is associated with restoration of alveolarization, enhanced angiogenesis and improvement in pulmonary vascular development. Intriguingly, miR-34a is also increased in RDS, evolving and established BPD patients, indicating its potential role in human neonatal lung injury. Pharmacologic miR-34a inhibition has clinical translational potential as a viable therapeutic option in the treatment of neonatal patients to prevent/ameliorate BPD.

## Methods
**Animals**. All experimental WT mice of the C57BL/6 strain were purchased from The Jackson Laboratory (Bar Harbor, ME) and housed in Yale or Drexel animal care facilities. In addition, *miR34a*−/− mice[71] and conditional *miR-34*[fl/fl][72] (JAX laboratory) and SPC-CreER (gift from Brigid Hogan, PhD, Duke University, USA) were housed in the Yale and Drexel Universities Animal Care Facilities (New Haven, CT and Philadelphia, PA, respectively). Mice were allowed free access to

standard food and water. All animal experiments proceeded in accordance with NIH policies and were approved by the Institutional Animal Care and Use Committees (IACUCs) at Yale and Drexel Universities. Both male and female newborn mice were used for the experiments, at specific postnatal ages, as noted in the figure legends.

**Cell culture**. T2AECs purification: T2AECs were freshly isolated from fetal mice (E19–E20) by the modified method of Wang et al.[73]. Briefly, mouse lungs were obtained from timed-pregnant C57B6 mouse (Jax labs, USA). The tissues were transferred into a 50 ml conical tube containing Dulbecco's modified Eagle's medium (DMEM) on ice and digested by pipetting up and down with collagenase.

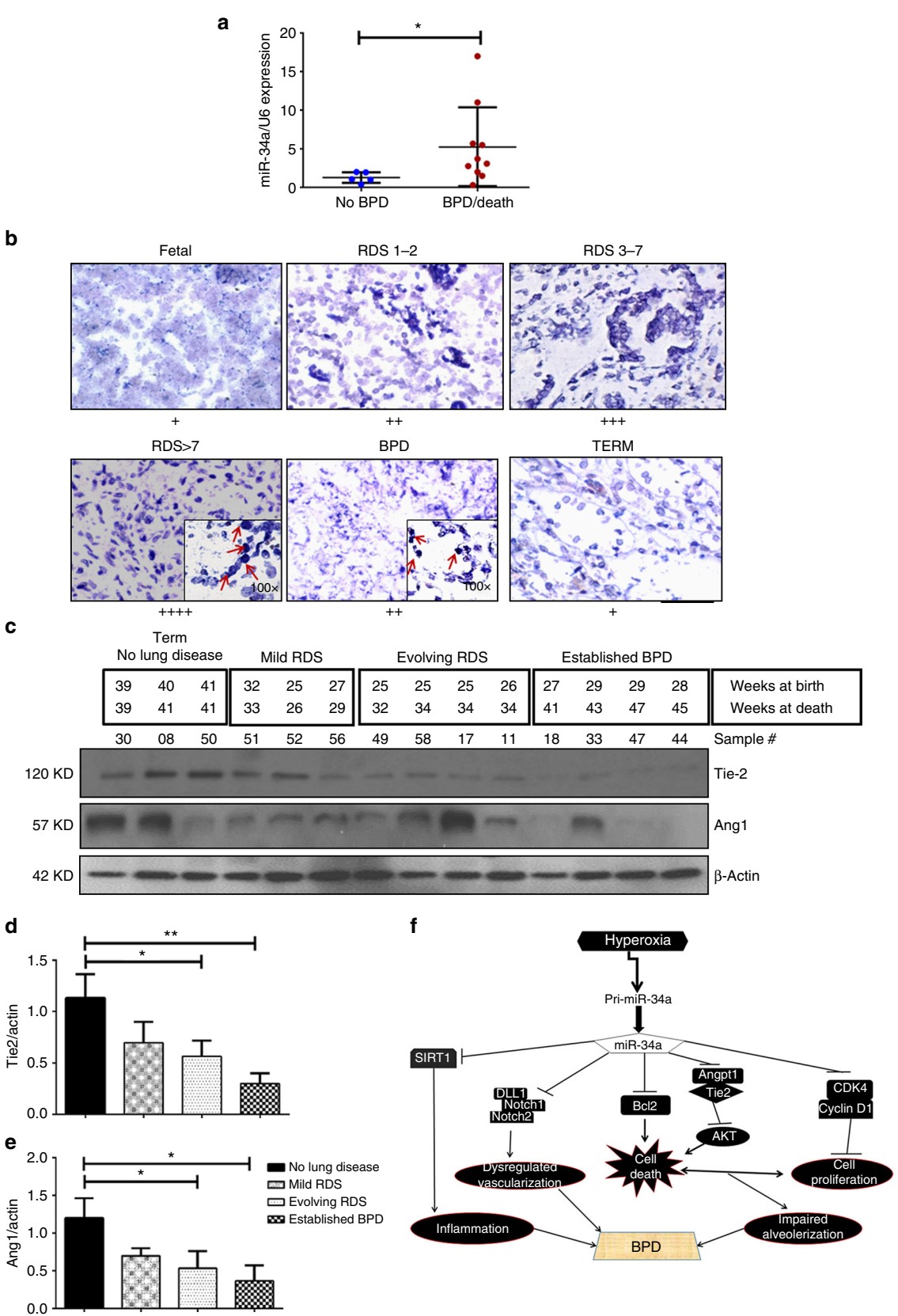

The cell suspension was serially filtered through 100, 30, and 20 µm nylon meshes using screen cups (Sigma). Clumped non-filtered cells from the 30 and 20 µm nylon meshes were collected after several washes with DMEM to facilitate the filtration of non-epithelial cells. Cells were next collected and based on epithelial cell morphology and immunostaining for SP-C (marker for type II cells), the purity of the cells was determined to be $90 \pm 5\%$ by microscopic analysis. The newly isolated T2AECs were further plated (till 50% confluency) on Bioflex multiwell plates (Flexcell International, Hillsborough, NC, USA) precoated with fibronectin. Monolayers were maintained in serum-free DMEM for further experimental analysis.

MLE12 cells were a gift from Patty J. Lee, MD (Yale University, USA), and were maintained in DMEM supplemented with 2% Fetal Bovine Serum (FBS), 100 U/ml penicillin, 100 µg/ml streptomycin (GIBCO) at 37 °C in 95% air; 5% $CO_2$. Hyperoxia conditions were achieved by placing 70–75% confluent MLE12 cells or freshly isolated T2AECs for specified time points in 95% $O_2$/5% $CO_2$ at 37 °C in a tightly sealed modular chamber (Stem Cell Technologies, Vancouver, Canada).

**Hyperoxia exposure**. Exposure to 100% $O_2$ to newborn (NB) mice (along with their mothers) was performed by placing them in cages in airtight Polypropylene chamber (medium size), following standard methods[74–77]. For hyperoxia experiments, all NB mice were exposed to 100% $O_2$ along with their mother for different time points. Every alternate day, the mother was changed with a lactating dam between RA and hyperoxia to prevent the mother from hyperoxia-induced death and to ensure sufficient nutrition for the treated pups during the course of hyperoxia exposure. We used two experimental lung injury models. In the HALI (hyperoxia-induced acute lung injury) model, exposure to 100% $O_2$ was initiated on postnatal day 1 (PN1; saccular stage of lung development) and continued till PN7 (alveolar stage). For the BPD model, NB mice were exposed to hyperoxia from PN1 to 4, then allowed to recover in RA for the following 10 days and then killed on PN14, as previously described[28,36,78]. Oxygen level was constantly monitored to be at 100%. The inside of the chamber was kept at atmospheric pressure, and mice were exposed to a 12 h light-dark cycle. $CO_2$ inhalation was used for euthanasia prior to removing lungs from the NB mice. In all experiments, NB mice were killed at selected time points and lungs were harvested for histology, RNA and protein evaluation. All animal work was approved by Yale and Drexel Universities IACUCs.

**Additional rodent models of BPD**. Hypoxia Model: Neonatal mice (C57BL/6 background strain) with an inducible dominant-negative mutation of the TGF-beta type II receptor (DNTGFbetaRII) and WT mice controls were exposed to hypoxia (12% $O_2$) or air from birth to 14 days of age following the method of Ambalavanan et al[39]. The DNTGFβRII mouse is a dominant-negative mutant as it expresses a cytoplasmically truncated TGFβRII receptor that competes with endogenous receptors for heterodimeric complex (TGFβRI and -RII) formation. The DNTGFβRII has no intrinsic activity as it lacks the cytoplasmic kinase domain, and the expression can be induced by administering $ZnSO_4$. This was administered (20 µg/g ip) daily to mice pups kept in air (DT-zinc-air group) or hypoxia (DT-zinc-hypoxia group). Control DNTGFβRII mice were administered saline (vehicle control) and kept in air (DT-saline-air group) or hypoxia (DT-saline-hypoxia group). Also, WT mice were administered saline and $ZnSO_4$ (same dose as mentioned above) as additional controls. RT-PCR was done to detect DNTGFβRII receptor mRNA using the primers: 5′-ATCGTCATCGTCTTTGTAGTC-3′ and 5′-TCCCACCGCACGTTCAGAAG-3′, to confirm induction of DNTGFβRII in the NB mice pups. No differences were noted in mortality of WT or DNTGFβRII mice (administered either vehicle or $ZnSO_4$) over the study duration. Standard techniques were utilized for collecting lungs of the mice pups after PN14 and isolating RNA from them, after completion of the study.

Chorioamnionitis: in a rat model of chorioamnionitis combined with PN hyperoxia exposure, 1 µg of lipopolysaccharide (LPS) was injected into each individual amniotic sac of the pregnant dams on E20 so as to induce chorioamnionitis on E21 and the pups to be normally delivered between E21 and E22. Briefly, a small incision was made in the abdomen of the pregnant dam on gestation day 20 following anesthesia, and carefully the pups were pulled out by lifting the uterine horn. Each individual amniotic sac was injected with 1 µg of LPS and the pups were placed back into the maternal abdominal cavity, the abdomen sutured and the mother was rested to deliver normally the following day. After birth, NB mice were exposed to hyperoxia (100% $O_2$) from PN1-7, and killed thereafter to obtain lung tissue. All animal work was approved by the University of Alabama at Birmingham and Thomas Jefferson University, Philadelphia IACUCs.

**Human lung tracheal aspirates**. Human lung tracheal aspirates (TA) pellets were obtained from premature infants being mechanically ventilated in the first PN week with an in-dwelling endotracheal tube. These infants had the final outcomes of having the diagnoses of with or without BPD and/or death. Collection and processing of the human lung samples was approved by the institutional review board of Yale University and Cooper University Hospital. Selected clinical details have been shown in Supplementary Table 1.

**Real-time reverse transcriptase PCR**. For the detection of miRNA expression, RNA was extracted from lungs, MLE12 cells, human tracheal aspirate (TA) pellets and primary T2AECs using miRNeasy mini kit (Qiagen, Valencia, CA). RNA concentration and quality was determined using a Biotek synergy II plate reader (Biotek, Winooski, VT). Across all samples the mean 260/280 ratio was greater than 2.0. cDNA was synthesized using a miScript II RT Kit (Qiagen, Valencia, CA). The StepOnePlus platform (Applied Biosystems) was used for all PCR, done in triplicate using miScript primer assay (Qiagen). Changes in expression were calculated by the change in threshold ($\Delta\Delta C_T$) method with RNU6 as the endogenous control for miRNA analysis and ß-actin (ACTB) for primary miRNA for gene-expression analysis. The miScript primer assay (Qiagen) IDs are mouse MS00001428 (miR-34a), Human MS00003318 (miR-34a), MS00033740 (RNU6), Mouse MP00005614 (Pre-miR-34a) and Mouse QT01136772 (ACTB).

**Western blot**. Western blotting was performed as previously described[79]. Briefly, lung lysate and whole-cells extracts were made in RIPA buffer and protein concentration determined by Bradford method (BioRad Dye). Proteins were separated by SDS-PAGE (4–20%) and transferred to PVDF or nitrocellulose (BioRad) membranes followed by 1 h blocking at room temperature (RT) in either Odyssey blocking Buffer (Licor, Germany) or 5% milk in TBST (Tris buffer saline with 0.1% Tween 20) and incubating with primary antibodies at 4 °C, overnight. The following day membranes were washed three times with either TBST or PBST (Phosphate buffer saline with 0.1% Tween 20), incubated with either HRP-conjugated or fluorescent conjugated secondary antibody, as and when necessary for 2 h at RT, washing and subsequently developing using enhanced chemiluminescence reagent (Amersham, Chalfont St Giles, UK) followed by development with autoradiography or LICOR infrared imaging.

The primary antibodies used were Ang1 (Millipore; AB10516; 1:1000), Ang2 (Millipore; AB10516; 1:1000), N-Cadherin (Millipore; AB10516; 1:1000), b-Catenin (Cell signalling; AB10516; 1:1000), Notch1 (Cell signaling; 3608; 1:1000), Notch2 (Cell signaling; 5732; 1:800), Acetylated P53 (Cell signaling; 2570; 1:500), P53 (Cell signaling; 2524; 1:2000), phospho-Tie2 (R&D; AF2720; 1:500), Tie2 (Abcam; ab24859; 1:1000), Sirt1 (Cell signaling; 2028; 1:800), cyclin D1 (Cell signaling; 2922; 1:800), phospho-c-Kit (Cell signaling; 3391; 1:800), c-Kit (Cell signaling; 3074; 1:1000), Stem cell factor (SCF) (Santa Cruz Biotechnology; SC-9132; 1:1000), Bcl2 (Cell signaling; 3498; 1:1000), CDK4 (Santa Cruz Biotechnology; SC-260; 1:1000), cleaved-caspase 3 (Cell signaling; 9661; 1:500), Caspase 3 (Cell signaling; 9662; 1:1000), phosphor-erk1/2 (Cell signaling; 9101; 1:1000), Total erk1/2 (Cell signaling; 9102; 1:1000), p-Akt (473) (Cell signaling; 9271; 1:1000), Akt (Cell signaling; 9272; 1:1000).

**Fig. 10** Lungs of infants with RDS and BPD have increased miR-34a expression. **a** miR-34a expression in cell pellets obtained from tracheal aspirates of neonates in the first PN week, who subsequently did or did not develop BPD. **b** Next, we used ISH to detect miR-34a in human neonatal lungs. As noted in the representative microphotographs, there was increased violet staining (miR-34a-positive) of the cells in the lungs of RDS and BPD neonates, compared to controls. **c** Western blot analysis of Tie2 and Ang1 was performed on total homogenates from human lung samples. **d, e** Densitometric analysis of Tie2 and Ang1 expression from infants born near term with no lung disease compared to near or post term with mild RDS, evolving BPD and established BPD. **f** A proposed schema for the role of miR-34a in the pathogenesis of BPD. Hyperoxia exposure to the developing lung leads to production and release of the primary (Pri-miR-34a), which is processed into the mature form of miR-34a. Downstream targets of the miR34a signaling pathway include Ang1 and its receptor Tie2, and the anti-apoptotic protein Bcl2; decreased expression of both are known to increase cell death in hyperoxia-induced lung injury models and BPD. In addition, hyperoxia decreases cell proliferation via CDK4 and cyclin D1, both targets of miR34a. The class III histone deacetylator, Sirt1 is also a downstream target of miR-34a, and a decrease in Sirt1 has been associated with enhanced transcription of pro-inflammatory mediators and BPD. The combined effect of enhanced cell death and decreased cell proliferation would be impaired alveolarization in the lung. In addition, miR34a, by suppressing the Ang1/Tie2 signaling pathway and enhancing cell death, results in dysregulated vascularization in the lung. Hence, increased miR-34a results in increased inflammation, impaired alveolarization and dysregulated vascularization in the developing lung—the hallmarks of "new" BPD. RDS: respiratory distress syndrome; BPD: bronchopulmonary dysplasia; Ang1: angiopoietin 1; Sirt1: Sirtuin 1. *$P < 0.05$, **$P < 0.01$, compared with controls, 1-way ANOVA

Equality of loading was confirmed by probing for β-actin (Santacruz, Cell signaling Technology, Danvers, MA) or GAPDH (Cell signaling Technology, Danvers, MA). The uncropped raw images of western blot using the above antibodies have been shown in Supplementary Fig. 10.

**Luciferase reporter assays**. Ang1 and Tie2 3′UTR reporter constructs for mouse were obtained from Genecopoeia along with control construct (Cmi T000001-MT01). All these targets were cloned in miRNA Target clone control vector for pEZX-MT01 (Genecopoeia). For luciferase assays, $5 \times 10^6$ MLE12 cells were transfected with endotoxin-free 5× 3′UTR Ang1 and Tie2 reporter luciferase plasmid (Genecopeia, Rockville, MD) and Luc-Pair miR Luciferase Assay Kit (Genecopeia). Cells were allowed to recover for 24 h before being transfected with these constructs as described above. Reporter gene activity was measured with the Dual-Luciferase kit (Promega) 24 h after hyperoxia treatment.

**Determination of cytokine and myeloperoxidase levels**. The levels of IL-6, and IL-1β in lung homogenates were measured by ELISA (R&D Systems). The lung myeloperoxidase (MPO) levels were determined using lung tissue homogenates using a mouse MPO ELISA kit (Catalog #ab155458; Abcam), according to manufacturer's instructions.

**Lung morphometry**. Alveolar size was estimated from the mean chord length of the airspace and septal thickness, as described previously, using ImageJ[76,77]. Briefly, hematoxylin-eosin sections (×100 magnification) were analyzed in ImageJ using the plugins and macros for chord length and septal thickness.

**TUNEL assay with T2AECs co-localization**. TUNEL assay was performed on paraffin lung sections (5 μm) using in situ Cell Death Detection Kit, Fluorescein (Roche) following manufacturer's instructions. Co-localization for T2AECs marker SP-C (surfactant protein C; Santacruz, 1:50) was done along with the apoptotic cells, as described[80]. Following TUNEL staining, the sections were incubated with SP-C antibody, overnight at 4 °C, quick washing in 1X PBS and incubated with fluorescent secondary antibody for 2 h at room temperature (Jackson immunoresearch, 1:200), subsequent washing with 1X PBS and mounting with DAPI (Vector labs, California). Quantification of TUNEL-positive cells co-expressing SPC was performed in selected images by an observer masked to the identity of the experimental groups.

**PAH-induced right ventricular hypertrophy**. Quantitative measurements of PAH-induced right ventricular hypertrophy (RVH) by RV/left ventricle (LV) and RV/(LV + interventricular septum or IVS) ratios were done using the methodology described previously either using ImageJ or Cell Sens Olympus software[31]. Briefly, the thickness of right and left ventricle was measured on hematoxylin-eosin sections (×40 magnification) and the ratio between the two regions of the heart were calculated.

**In situ hybridization for human lung samples**. Human lung tissue samples were obtained postmortem from premature infants having the diagnoses of RDS: 1–2 days (RDS 1–2), RDS 3–7 days (RDS 3–7), RDS >7 days (RDS >7), BPD. Collection and processing of the human lung samples was approved by the National Supervisory Authority for Welfare and Health in Finland and the University of Rochester Institutional Review Board. Selected clinical details have been shown in Supplementary Table 2. Whole lungs from humans were isolated and immediately fixed with 10% NBF (neutral-buffered formalin). Briefly, lung sections were subjected to deparaffinization, incubation with 0.5% pepsin solution (20 min at 37 °C in humidified chamber), dehydration, and hybridization with either 40 nM Biotin LNA miR-34a probe, at 55 °C for 3 h. Subsequently sections were washed, blocked and incubated with streptavidin-AP reagent for 20 min and applied with alkaline phosphatase solution containing nitro-blue tetrazolium and 5-bromo-4-chloro-3=indolyphosphate (BCIP/NBT) for 1 h. Finally sections were dehydrated, mounted, and examined under microscope. Counterstaining was omitted for clarity.

**Prediction and identification of miRNA gene targets**. To identify potential targets for differentially expressed miRNAs, we screened their sequences against the mouse genome database, using the miRNA target identification programs miRBase, PicTar, and TargetScan.

**Flow cytometry**. Briefly, after the hyperoxia exposure and treatment, MLE12 cells were trypsinized and washed with cold PBS. Following this, cells were instantly stained using FITC Annexin V Apoptosis Detection Kit (BD Pharmingen) according to the manufacturer's protocol. Cell density was determined and stained with Annexin V and PI and analyzed by flow cytometry (Becton Dickinson). A worklist was created from the assay and the samples were acquired automatically using the Loader with acquisition criteria of 10,000 events for each tube. The report generated from the apoptosis assay included the following gates and plots:
1. FSC-A vs. SSC-A with a gate for cells.

2. Annexin V FITC-A vs propidium iodide-A (PI-A) with gates for following populations: (1) Annexin V–/PI– (2) Annexin V+/PI– (3) Annexin V+/PI+ (4) Annexin V–/PI+.

A summary of assay results with statistics for untreated and treated samples was automatically calculated.

**Statistical analyses**. Values are expressed as mean ± SEM. Groups were compared with the Student's two-tailed unpaired $t$-test or 1-way ANOVA (followed by Tukey's Multiple Comparison post hoc test) or 2-way ANOVA as appropriate (followed by Bonferroni's Multiple Comparison post hoc test), using GraphPad Prism 7.0 (GraphPad Software, Inc., San Diego, CA). A value of $p < 0.05$ was considered statistically significant.

**Data availability**. All relevant data are available from the authors upon reasonable request.

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

## Acknowledgements

Supported in part by grants from The Hartwell Foundation (M.S., P.D., and V.B.), HL63039 (G.P.) from the NHLBI of the National Institutes of Health, and Sigrid Jusélius Foundation (S.A.).

## Author contributions

Concept and design: M.S. and V.B. Acquisition of data: M.S., P.D., A.P., Z.H.A., A.K., Z. W.Z., N.A., G.P., S.A., and V.B. Data analysis and interpretation: M.S., P.D., A.P., Z.W.Z., N.A., and V.B. Drafting and/or critical revision for intellectual content: M.S., P.D., A.P., Z.H.A., A.K., Z.W.Z., N.A., G.P., S.A., and V.B.

## Additional information

**Competing interests:** The authors declare no competing financial interests.

