## [Peer Review file · Nature Communications]

Reviewers' comments:

Reviewer #1 (Remarks to the Author):

Bronchopulmonary dysplasia (BPD) is the most common chronic lung disease in infants and has very significant short- and long-term consequences for those affected. There is currently no specific preventive or therapeutic agent to alleviate BPD. Thus, the goal of this study to identify new targetable molecular mechanisms operating in the disease process is important. The hypothesis that miR-34a is specifically involved in the pathogenesis of this disease has not been previously explored. The authors of the present study carried out an extensive series of studies in vivo and in vitro to conclusively show that miR-34a contributes at least to neonatal murine BPD by influencing significantly T2AEC apoptosis through the regulation of Ang1-Tie2 signaling. The Reviewer agrees that these are novel findings. The fact that hyperoxia induces miR34 expression in T2AECs of newborn mice is clear and well supported. Whether the increased expression of miR-34a in human infants is due simply to hyperoxia is very unclear and not supported by the data. Very importantly, using genetic gain and loss of function strategies, the authors comprehensively prove a causal detrimental role of increased miR-34a in the development of hyperoxia-induced murine BPD. Experiments also show that inhibition of miR-34a is protective in both BPD and pulmonary and associated PAH phenotypes. Lastly, the investigators have documented important downstream targets of miR-34a, which are involved in angiogenic, inflammatory, cell death and cell proliferation pathways. They present good evidence that Ang1 and Tie2 are involved and that Ang1 treatment is protective.

Specific Comments:

1. Currently, lacking is any insight into the mechanisms that specifically control the increased expression of miR-34a. The authors raise the possibility, which has been established by others, that there is a reciprocal feed-forward loop between miR-34a and P53. Data demonstrating existence of this loop is not well documented in either the hypoxia or TGF β model of BPD included in the manuscript. The P53 data in the hyperoxia model is not highly supportive of a role for P53 and is less convincing than the less well-studied TGF β model presented in Fig. 11. It would seem important to at least begin to understand why miR-34a is increased (while others miRs of importance to the disease process are decreased) in the setting of hyperoxic lung injury. Further, to generalize miR-34a to the human studies, it would seem that there should be an explanation observed across all models (including hypoxia and chorioamnionitis) that lead to a BPD-like phenotype in rodents specifically leading to upregulation of miR-34a in the T2AEC.
2. The findings that targeting miR-34a signaling also abrogates the vascular abnormalities often observed in the animal model and human infants with BPD is interesting and important. It is a little unclear as to how exactly this might be achieved. Is miR-34a excreted from the epithelial cell? Does it have effects on endothelial cells, fibroblasts, macrophages, etc.? The question is raised in part by the observation that miR-34a is increased in the tracheal aspirate of patients with BPD. Is this simply a reflection that there were sloughed epithelial cells in the tracheal aspirate or is there exosomal miR-34a in the BAL, which could be having paracrine effects?

Reviewer #2 (Remarks to the Author):

There are currently no effective treatments for bronchopulmonary dysplasia (BPD). In this manuscript Syed et al assess the potential of targeting MiR-34a function for inhibiting disease inducing hyperoxia in a mouse models of BPD. Although targeting miRNA function may prove to be difficult in humans this study importantly demonstrates the role of this small non-coding RNA in pathogenesis and highlights the potential of targeting miR-34a, or importantly pathways it regulates, as potential new therapeutic approaches.

In this investigations Syed et al demonstrate that in neonatal mice lungs exposed to

hyperoxia miR-34a levels are significantly increased and this is directly associated with increased cell death and cellular proliferation. Inhibition of miR-34a function globally and in type 2 alveolar epithelial cells (T2AECs) attenuates cell death and inflammation with injury. This effect was associated by modulation of the levels of factors (e.g. Ang1, Tie2, Notch2, caspases, SCF and IL-1 beta) that regulate inflammation, cell death and proliferation and angiogenesis. This was observed in both hyperoxia-induced acute lung injury (HALI) and BPD models. Overexpression of miR-34a exacerbated clinical/pathological features of both BPD pulmonary and PAH. Importantly, the investigators demonstrate that angiopoietin-1, a target of this miR, when administered to the lung attenuated the development of both BPD pulmonary and PAH. The data in the mouse model was then translated to human disease where an association between increased miR-34a levels and localization to T2AECs from neonates with respiratory distress syndrome (RDS) and BPD was demonstrated.

This is a well-controlled and written study. It will make an excellent contribution to the field.

For consideration:

Fig 1. miR-34a levels do not appear to be significantly increased in BPD? "even the BPD model showed a significant increase in miR-34a expression as compared to RA control (Fig. 1A)". This difference is small as is the sample size and more replicates are required.

"expression of miR-34a was highest with 95% O₂ exposure at 24h (Fig. 1D) and with 60% O₂ exposure at 48h (Fig. 1E)." How do these levels of exposure relate to clinical application?

Fig S1- I cannot read the axis labels of (A). The legend does not describe the fig properly. Where is NB? = PN? (B-C) State the age of the mice where these cells were isolated? Why only n=1 for (A) and 3 animals per group? What is the reproducibility? n=4 stated in Fig 1. Please clarify.

"MiR-34a miRNA was detected in lungs from WT mice breathing RA and increased markedly after exposure to 100% O₂ (Supplementary Fig. 1A)"- from 1 mouse?

"miR-34a expression is increased upon hyperoxia exposure in developing lungs, and this appears to be localized to T2AECs" Should state of the cells investigated (only 3 of many cell types in the lung!). Otherwise this is an overstatement and misleading.

Fig. 2 A. Can you provide quantitative evidence for your Western blots (densitometry) (C-E)? This would greatly advance the significance of these observations.

Fig. 4. Once again can (C) be quantified to demonstrate significance.

Fig. 5 (A) very difficult to read some axis.

Reviewer #3 (Remarks to the Author):

Some suggestions for the authors to consider.

1. To consider the full picture of Ang/Tie2 axis, it may be important to look for Ang 2 expression in this model/
2. Authors should consider infection model of BPD to see miR-34a is altered; this may be more relevant for the human situation.
3. What is the common sequence in the 3'UTR of ang1 and Tie 2 that miR-34a binds. This should

be included a sub-panel.

4. miR-34a regulates EMT. Is there a role for this process in BPD?

5. Summary figure of the miR-34a in the pathogenesis of BPD would be nice for readers

Reviewer 1:

Bronchopulmonary dysplasia (BPD) is the most common chronic lung disease in infants and has very significant short- and long-term consequences for those affected. There is currently no specific preventive or therapeutic agent to alleviate BPD. Thus, the goal of this study to identify new targetable molecular mechanisms operating in the disease process is important. The hypothesis that miR-34a is specifically involved in the pathogenesis of this disease has not been previously explored. The authors of the present study carried out an extensive series of studies in vivo and in vitro to conclusively show that miR-34a contributes at least to neonatal murine BPD by influencing significantly T2AEC apoptosis through the regulation of Ang1-Tie2 signaling. The Reviewer agrees that these are novel findings. The fact that hyperoxia induces miR34 expression in T2AECs of newborn mice is clear and well supported. Whether the increased expression of miR-34a in human infants is due simply to hyperoxia is very unclear and not supported by the data. Very importantly, using genetic gain and loss of function strategies, the authors comprehensively prove a causal detrimental role of increased miR-34a in the development of hyperoxia-induced murine BPD. Experiments also show that inhibition of miR-34a is protective in both BPD and pulmonary and associated PAH phenotypes. Lastly, the investigators have documented important downstream targets of miR-34a, which are involved in angiogenic, inflammatory, cell death and cell proliferation pathways. They present good evidence that Ang1 and Tie2 are involved and that Ang1 treatment is protective.

Specific Comments:

C1. *Currently, lacking is any insight into the mechanisms that specifically control the increased expression of miR-34a. The authors raise the possibility, which has been established by others, that there is a reciprocal feed-forward loop between miR-34a and P53. Data demonstrating existence of this loop is not well documented in either the hypoxia or TGF β model of BPD included in the manuscript. The P53 data in the hyperoxia model is not highly supportive of a role for P53 and is less convincing than the less well-studied TGF β model presented in Fig. 11. It would seem important to at least begin to understand why miR-34a is increased (while others miRs of importance to the disease process are decreased) in the setting of hyperoxic lung injury. Further, to generalize miR-34a to the human studies, it would seem that there should be an explanation observed across all models (including hypoxia and chorioamnionitis) that lead to a BPD-like phenotype in rodents specifically leading to upregulation of miR-34a in the T2AEC.*

R1: We would like to thank the reviewer for the positive comments regarding the comprehensive nature and the novelty of the data shown in our manuscript. We investigated the potential connection with p53 with miR34a, given the previous reports, as noted by the reviewer, too. As noted in **Supplemental Figs. 2B and 2C**, we found a modest (non-significant) decrease in miR34a expression, upon exposure to hyperoxia *in vitro* and *in vivo*. In addition, we have now evaluated miR34a expression in p53 null mutant and Trp53 siRNA treated mice in room air and our BPD model at PN14. These data are shown in **Supplemental Figs. 2D-E**, where miR34a expression is significantly increased in room air, but significantly decreased in the BPD model, upon p53 inhibition.

We further investigated the role of hypoxia and TGF β signaling in a newborn mice model (PMID: 18487357). We obtained lung tissues and noted that miR34a expression was decreased with hypoxia and decreased TGF β signaling [using inducible dominant-negative mutation of the TGF-beta type II receptor (DNTGFbetaR2) mice] or a combination of the two exposures (**Fig. 11C**). Furthermore, we used antenatal LPS administration (mimicking chorioamnionitis) with/without additional PN hyperoxia exposure in a neonatal rat model, and noted increased expression of miR34a in the lungs, but only when hyperoxia exposure was present (**Fig. 11D**). While we cannot state that the increase in miR34a expression in the lungs of human infants is exclusively due to hyperoxia exposure, we do believe that hyperoxia is a major contributor to the increased levels of miR34a, as noted by the novel data provided in 4 additional rodent models of hyperoxia-induced lung injury/BPD (**Supplemental Figs. 2D-E, Figs. 11C-D**) in the *revised* manuscript.

C2. *The findings that targeting miR-34a signaling also abrogates the vascular abnormalities often observed in the animal model and human infants with BPD is interesting and important. It is a little unclear as to how exactly this might be achieved. Is miR-34a excreted from the epithelial cell? Does it have effects on endothelial cells, fibroblasts, macrophages, etc.? The question is raised in part by the observation that miR-34a is increased in the tracheal aspirate of patients with BPD. Is this simply a reflection that there were sloughed epithelial cells in the tracheal aspirate or is there exosomal miR-34a in the BAL, which could be having paracrine effects?*

R2: Tracheal aspirates obtained from human lungs usually contain a mixture of epithelial and inflammatory cells (neutrophils and mononuclear cells/macrophages), as reported by us (PMID: 22273724), and other investigators (PMIDs: 3769730, 1601013, 9570027, 18353056). We have checked the expression of miR34a in endothelial cells and macrophages exposed to hyperoxia, and have not noted any changes in miR34a levels. Our data would suggest that miR34a is released from epithelial cells, specifically Type II pneumocytes, and this is the source of miR34a in the tracheal aspirates. We are not aware of data if miR34a is packaged and released from epithelial cells via exosomes in the BAL.

Reviewer 2:

There are currently no effective treatments for bronchopulmonary dysplasia (BPD). In this manuscript Syed et al assess the potential of targeting MiR-34a function for inhibiting disease inducing hyperoxia in a mouse models of BPD. Although targeting miRNA function may prove to be difficult in humans this study importantly demonstrates the role of this small non-coding RNA in pathogenesis and highlights the potential of targeting miR-34a, or importantly pathways it regulates, as potential new therapeutic approaches.

In this investigations Syed et al demonstrate that in neonatal mice lungs exposed to hyperoxia miR-34a levels are significantly increased and this is directly associated with increased cell death and cellular proliferation. Inhibition of miR-34a function globally and in type 2 alveolar epithelial cells (T2AECs) attenuates cell death and inflammation with injury. This effect was associated by modulation of the levels of factors (e.g. Ang1, Tie2, Notch2, caspases, SCF and

IL-1 beta) that regulate inflammation, cell death and proliferation and angiogenesis. This was observed in both hyperoxia-induced acute lung injury (HALI) and BPD models. Overexpression of miR-34a exacerbated clinical/pathological features of both BPD pulmonary and PAH. Importantly, the investigators demonstrate that angiotensin-1, a target of this miR, when administered to the lung attenuated the development of both BPD pulmonary and PAH. The data in the mouse model was then translated to human disease where an association between increased miR-34a levels and localization to T2AECs from neonates with respiratory distress syndrome (RDS) and BPD was demonstrated.

This is a well-controlled and written study. It will make an excellent contribution to the field.

Minor Comments:

C1. *Fig 1. miR-34a levels do not appear to be significantly increased in BPD? “even the BPD model showed a significant increase in miR-34a expression as compared to RA control (Fig. 1A)”. This difference is small as is the sample size and more replicates are required.*

R1: We thank the reviewer for the positive comments regarding our manuscript.

As recommended, we have added more replicates and a separate figure as **Supplementary Fig. 1B**, in the *revised* manuscript.

C2. *“expression of miR-34a was highest with 95% O₂ exposure at 24h (Fig. 1D) and with 60% O₂ exposure at 48h (Fig. 1E).” How do these levels of exposure relate to clinical application?*

R2: In the clinical scenario, most preterm neonates are usually exposed to higher concentrations of supplemental O₂ initially, with an attempt to decrease the hyperoxia exposure on subsequent days. Our data suggest that both the concentration and duration of hyperoxia exposure increases the expression of miR34a in the lungs. Hence, a lower concentration of supplemental O₂ for a longer duration would also increase miR34a, mimicking the clinical situation.

C3. *Fig S1- I cannot read the axis labels of (A). The legend does not describe the fig properly. Where is NB? = PN? (B-C) State the age of the mice where these cells were isolated? Why only n=1 for (A) and 3 animals per group? What is the reproducibility? n=4 stated in Fig 1. Please clarify.*

R3: We have now made the axis label of Fig. S1A more legible. We have removed the abbreviation “NB”. We retained the abbreviation and expanded form of “PN”, as it is mentioned in the figure legend. The age of the mice was PN4 and this has now been stated in the figure legend. The n=1 referred to the single microarray analysis done. We have deleted it to avoid confusion. We used n=3 mice per group for the microarray analysis. We have increased the replicates in Fig. 1, as stated above.

C4. *MiR-34a miRNA was detected in lungs from WT mice breathing RA and increased markedly after exposure to 100% O₂ (Supplementary Fig. 1A)”- from 1 mouse?*

R4: The n=1 referred to the single microarray analysis done. We have deleted it to avoid confusion. We used n=3 mice per group for the microarray analysis. We performed one miRNA microarray analysis and chose 5-6 miRNAs for validation and miR-34 was the most increased miRNA among them. We also performed the time course experiments and found consistent increase in miR-34a only.

C5. *“miR-34a expression is increased upon hyperoxia exposure in developing lungs, and this appears to be localized to T2AECs” Should state of the cells investigated (only 3 of many cell types in the lung!). Otherwise this is an overstatement and misleading.*

R5: Thank you for pointing that out. We have corrected that statement in the revised manuscript.

C6. *Fig. 2 A. Can you provide quantitative evidence for your Western blots (densitometry) (C-E)? This would greatly advance the significance of these observations.*

R6: We have now provided quantification (densitometric data) for **Figs. 2C-E**, and shown as **Figs. 2C-H** in the revised manuscript.

C7. *Fig. 4. Once again can (C) be quantified to demonstrate significance.*

R7: As suggested, we have provided quantification of Tie2 and Ang1 in **Figs. 4E** and **4F**, in the revised manuscript.

C8. *Fig. 5 (A) very difficult to read some axis.*

R8: We have now made the axis labels of Fig. 5A more legible, in the revised manuscript.

Reviewer 3:

Some suggestions for the authors to consider.

Specific Points:

C1. *To consider the full picture of Ang/Tie2 axis, it may be important to look for Ang 2 expression in this model.*

R1: We have now shown Ang2 protein expression levels in **Figs. 5J** and **5K** upon hyperoxia exposure at PN4 and in the BPD model at PN14 in WT and miR34a null mutant mice lungs.

C2. *Authors should consider infection model of BPD to see miR-34a is altered; this may be more relevant for the human situation.*

R2: We used an antenatal LPS exposure (mimicking chorioamnionitis) with/without additional hyperoxia exposure model, and noted increased expression of miR34a in the lungs, only when hyperoxia exposure was present (**Fig. 11D**).

C3. *What is the common sequence in the 3'UTR of ang1 and Tie 2 that miR-34a binds. This should be included a sub-panel.*

R3: As recommended, we show the common sequence in a sub-panel as **Fig. S3A**.

C4. *miR-34a regulates EMT. Is there a role for this process in BPD?*

R4: We evaluated 2 EMT markers (N- and E-cadherin) in lung homogenates of WT and miR-34a null mutant mice upon hyperoxia exposure at PN4 and in the BPD model at PN14. This has now been shown in **Fig. S8**. We noted that the mesenchymal marker (N-cadherin) is decreased in the miR-34a null mutant mice, as compared to WT, in room air, PN4 hyperoxia and BPD lungs at PN14, with no change in the epithelial marker (E-cadherin). Overall, our results would suggest that there is a potential role for EMT in the BPD model, but further experiments (beyond the scope of this manuscript) would be required to be definitive.

C5. *Summary figure of the miR-34a in the pathogenesis of BPD would be nice for readers*

R5: We thank the reviewer for this suggestion. We included a schematic diagram (**Fig. 12F**) and corresponding legend in the *revised* manuscript.

REVIEWERS' COMMENTS:

Reviewer #1 (Remarks to the Author):

the authors have addressed the previous issues raised by this reviewer.
the manuscript is well done and presents exciting new science.
i believe that the quality of some of the figures could be improved as it would complement the science presented.

Reviewer #2 (Remarks to the Author):

Comments addressed

Reviewer #3 (Remarks to the Author):

I have no further comments

Reviewer 1:

C1. *the authors have addressed the previous issues raised by this reviewer. the manuscript is well done and presents exciting new science. i believe that the quality of some of the figures could be improved as it would complement the science presented..*

R1: We would like to thank the reviewer for the positive comments regarding our response and the novelty of the data shown in our manuscript. We have done our best with the figures.

Reviewer 2:

C1. *Comments addressed.*

R1: We thank the reviewer for the positive comments regarding our response.

Reviewer 3:

C1. *I have no further comments.*

R1: We thank the reviewer for the positive comments regarding our response.